# LLM-Guided Search for Deletion-Correcting Codes

**Franziska Weindel**                                                    *franziska.weindel@tum.de*
*Technical University of Munich*
*Munich Center for Machine Learning*

**Reinhard Heckel**[*]                                                   *reinhard.heckel@tum.de*
*Technical University of Munich*
*Munich Center for Machine Learning*

**Reviewed on OpenReview:** *https://openreview.net/forum?id=qZ69Ozpo6v*

## Abstract

Finding deletion-correcting codes of maximum size has been an open problem for over 70 years, even for a single deletion. We adapt FunSearch, a large language model (LLM)-guided evolutionary search, to discover functions that construct deletion-correcting codes at short code lengths. For a single deletion, our search finds a function that we prove constructs the conjectured-optimal Varshamov-Tenengolts code. For multiple deletions and quaternary edit codes, the discovered functions improve on prior explicit, search-based, and neural constructions but remain empirical heuristics without new theoretical insights. We study design choices for LLM-guided evolutionary search and find that, for our problem, compute is better allocated to sampling more functions than to longer reasoning traces per function, and that co-evolving natural language descriptions with code hurts search quality. We propose deduplicating logically identical functions during evolution, which we find critical for search diversity. Our results demonstrate the potential of LLM-guided evolutionary search for information theory and code design and represent the first application of such methods for constructing error-correcting codes. However, in our current formulation, evaluating a function scales exponentially with code length, limiting the approach to short codes.

## 1 Introduction

Large language models (LLMs) are increasingly used for scientific discovery and problem solving, in particular for mathematical and algorithmic problems (Fawzi et al., 2022; Mankowitz et al., 2023; Guo et al., 2025b). An important class of methods for discovering new algorithms and mathematics is LLM-guided evolutionary search (Romera-Paredes et al., 2024; Novikov et al., 2025; Lange et al., 2025; Yu et al., 2025; Assumpção et al., 2025). Romera-Paredes et al. (2024) demonstrate that LLM-guided evolutionary search can discover new algorithmic solutions for problems that are difficult to solve but easy to evaluate. Their method, FunSearch (Function Space Search), represents combinatorial problems as Python code and searches for algorithmic solutions, improving on the best-known results for the cap set and online bin-packing problems.

In this paper, we adapt FunSearch to find new error-correcting codes at short code lengths and study FunSearch variants to see which design choices enable solving algorithmic problems effectively.

Error-correcting codes enable reliable communication and data recovery from storage media, even in the presence of errors and defects. In a typical coding scheme, an encoder maps information to a codeword, which is corrupted by errors during transmission, and a decoder attempts to recover the original message. While substitutions and erasures are well understood, deletions are significantly more challenging. Deletions shift subsequent symbols, disrupting the memoryless property typically assumed in coding theory.

---

[*]Correspondence to: `reinhard.heckel@tum.de`

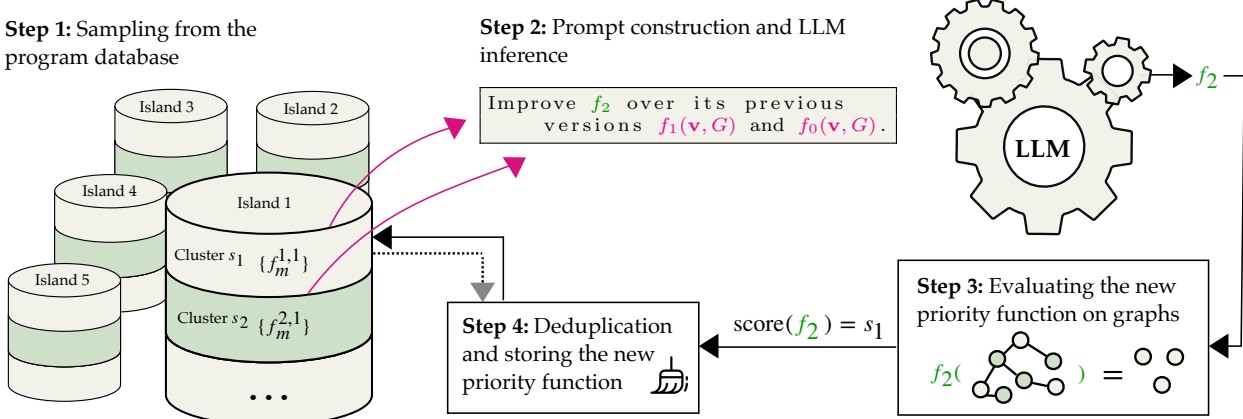

Figure 1: FunSearch for finding error-correcting codes at short lengths. Each iteration samples few-shot examples from the database, asks the LLM to generate an improved priority function, and evaluates it by greedily constructing error-correcting codes. Executable, non-duplicate functions are added to the database.

For a fixed number of correctable errors, better codes have larger code sizes. Despite significant effort, determining the maximum code size for a fixed number of adversarial deletions has proven difficult using traditional human-driven approaches to information theory. A class of codes known as Varshamov-Tenengolts (VT) codes (Varshamov & Tenengolts, 1965) achieves the maximum code size for correcting a single deletion as the code length approaches infinity (Levenshtein, 1966). However, for finite code lengths, the gap to the upper bound on code size is large even at moderate lengths (Kulkarni & Kiyavash, 2013). Although VT codes are conjectured to be maximum-size for all lengths and a single deletion, their optimality has only been proven for lengths up to 11 (Butenko et al., 2002; Nakasho et al., 2023; Sloane, 2002).

Given the limited progress with traditional techniques, LLM-guided evolutionary search offers a new approach to constructing deletion-correcting codes. We adapt FunSearch to generate priority functions that greedily construct codes by assigning priorities to sequences and adding the highest-priority sequence while ensuring error-correcting constraints are satisfied. We focus on binary deletion-correcting codes, where many fundamental questions remain open (Sloane, 2002), and additionally apply it to quaternary insertion-deletion-substitution (edit) codes, motivated by applications in biological sequencing and DNA data storage (Bar-Lev et al., 2024).

While FunSearch relies on sampling many functions, viewing the LLM as a source of diversity with occasionally useful ideas, subsequent work focuses on investing more compute per function (Liu et al., 2024; Ye et al., 2024; Surina et al., 2025). Under a fixed inference budget, this reduces the number of functions sampled, raising the question of whether discovery comes from sampling many functions or spending more compute per function. We study this trade-off by varying how much the LLM reasons before generating a new priority function and find that for our problem, compute is better allocated to sampling more functions than to longer reasoning traces.

Our main contributions are:

- We provide a proof of concept that LLM-guided evolutionary search is a promising tool for discovering error-correcting codes. For a single deletion, our search rediscovers a known optimal construction: it finds a function that we prove constructs the conjectured-optimal VT code for any code length. For multiple deletions and quaternary edit codes, our functions outperform explicit constructions (Helberg & Ferreira, 2002; Abdel-Ghaffar et al., 2011), search-based methods (Landjev & Haralambiev, 2007; Swart & Ferreira, 2003), and a neural method (Guo et al., 2025a). However, they do not match the best-known sizes from general-purpose graph solvers at longer lengths, and analyzing them to derive new coding-theoretic insights or efficient encoders remains an open problem.

- We study design choices for LLM-guided evolutionary search, finding that, for our problem, sampling more functions is more effective than longer reasoning traces per function, and that co-evolving

natural language descriptions with code hurts search quality. Moreover, we propose to deduplicate logically identical functions during evolution, which we find critical, as without it we could not find any function matching conjectured-optimal code sizes.

- We provide an implementation that scales across multiple nodes via asynchronous message passing and uses batched LLM inference with locally hosted models, unlike existing implementations (Sharma, 2025; Ye et al., 2024; Assumpção et al., 2025; Liu et al., 2024) that parallelize on a single node with API-based inference. We release our code[1] to facilitate future discovery by LLM-guided evolutionary search.

Our results demonstrate the potential of LLM-guided evolutionary search for information theory and code design. However, our approach is currently limited to short code lengths as evaluating priority functions scales exponentially with code length. The discovered functions cannot yet construct standalone codes at longer lengths but can be used as inner codes.

## 2 Related work

We review related work on LLM-guided evolutionary search and deletion-correcting codes.

### 2.1 Related work on LLM-guided evolutionary search

As mentioned, our work builds on existing approaches to LLM-guided evolutionary search, in particular FunSearch (Romera-Paredes et al., 2024). Lehman et al. (2023) first demonstrate the effectiveness of using LLMs to aid evolutionary algorithms, using the LLM as an intelligent mutator for automatic data generation. AlphaEvolve (Novikov et al., 2025) extends FunSearch by evolving entire codebases rather than individual functions. Other applications of LLM-guided evolutionary search include neural architecture search (Zheng et al., 2023; Liu et al., 2025; Chen et al., 2023; Nasir et al., 2024), prompt optimization (Yang et al., 2023; Fernando et al., 2024), reward design for reinforcement learning (Ma et al., 2023; Hazra et al., 2024), scientific discovery in mathematics (Georgiev et al., 2025), and systems research (Cheng et al., 2025).

The most relevant application for finding error-correcting codes is heuristic design for combinatorial optimization. Liu et al. (2024) propose EoH, which improves performance and sample efficiency over FunSearch by evolving both algorithmic summaries and code jointly. Ye et al. (2024) introduce ReEvo, which incorporates reflection into the search by prompting the LLM to compare previously generated solutions. Dat et al. (2024) propose two diversity metrics and find that FunSearch and ReEvo stagnate in local optima due to low diversity, while EoH trades off diversity for performance.

### 2.2 Related work on deletion-correcting codes

The best-known single-deletion-correcting codes are VT codes (Varshamov & Tenengolts, 1965). For code length $n$ and parameter $a \in \mathbb{Z}$, the $\mathrm{VT}_a(n)$ code is defined as the set of binary sequences $\mathbf{v} \in \{0, 1\}^n$ satisfying $\sum_{i=1}^{n} iv_i \equiv a \pmod{n+1}$, where each bit $v_i$ is weighted by its position $i$ in the sequence. Levenshtein (1966) proves that the $\mathrm{VT}_0(n)$ code is asymptotically optimal and proposes a linear-time decoding algorithm. The $\mathrm{VT}_0(n)$ code is also conjectured to be the largest for all finite lengths, but this has only been proven for lengths up to $n = 11$ (Sloane, 2002; Butenko et al., 2002; Nakasho et al., 2023).

For multiple deletions $s > 1$, the optimal redundancy lies between $s \log n$ and $2s \log n$ (Levenshtein, 2002). Several constructions achieve $O(s \log n)$ redundancy asymptotically (Brakensiek et al., 2018; Gabrys & Sala, 2019; Sima et al., 2020b; Sima & Bruck, 2021; Sima et al., 2020a; Guruswami & Håstad, 2021). These constructions partition the codeword into short blocks, with block boundaries and contents protected separately. For block contents, they use short deletion-correcting codes built via greedy construction. We show that our discovered priority functions construct larger codes than random greedy and can therefore be used as inner codes in these constructions to improve redundancy.

---

[1]https://github.com/MLI-lab/DistributedFunSearch

```
"""
Finds large independent set in graph G where nodes are binary strings of length n.
Nodes in G are connected if they share a subsequence of length at least n - s.
Improve priority_v1 over its previous versions below.
"""
import numpy as np
import networkx as nx

def priority_v0(node, G, n, s):
    """Returns the priority with which we want to add node. Higher priority nodes are added
    first."""
    return 0.0

def priority_v1(node, G, n, s):
    """Improved version of priority_v0"""
```

Figure 2: Initial prompt.

For finite-length multiple-deletion-correcting codes, Helberg & Ferreira (2002) extend VT codes with an explicit construction that Abdel-Ghaffar et al. (2011) later proved corrects multiple deletions. However, the resulting code sizes are small at longer lengths. Swart & Ferreira (2003) find larger codes for two deletions and lengths $n \leq 12$ by greedy search over random permutations in a run-length representation. Similarly, Landjev & Haralambiev (2007) search in the space of all binary sequences, constructing codes for $s = 2, 3, 4, 5$ deletions and lengths $n \leq 30$.

## 3 Problem statement

We consider the problem of constructing large $n$-bit, $s$-deletion-correcting codes.

A deletion-correcting code is a set of sequences such that, even if an adversary deletes $s$ bits from a sequence, the original sequence can still be uniquely recovered. A subsequence of length $n - s$ is obtained by deleting $s$ bits while preserving the order of the remaining bits. Unique recovery is not possible if two sequences share a common subsequence of length at least $n - s$. Thus, an $n$-bit, $s$-deletion-correcting code is a set $\mathcal{C} \subseteq \{0, 1\}^n$ such that no two distinct codewords $\mathbf{v}, \mathbf{v}' \in \mathcal{C}$ share a common subsequence of length $n - s$.

The problem of constructing $n$-bit, $s$-deletion-correcting codes can be reduced to finding an independent set in a graph $G$ defined as follows (Shannon, 2003). Let $G$ be an undirected graph where each node is a binary sequence of length $n$, and two nodes are connected by an edge if and only if they share a common subsequence of length at least $n - s$. An independent set $\mathcal{I}$ in $G$ is a subset of nodes such that no two are connected by an edge, and any such independent set forms an $n$-bit, $s$-deletion-correcting code. Each choice of code length $n$ and deletion count $s$ defines a problem instance with a corresponding graph $G$.

To construct deletion-correcting codes, we greedily build independent sets $\mathcal{I}$ in $G$. Let $f(\mathbf{v}, G)$ be a priority function that assigns a real-valued priority to each node $\mathbf{v}$. At each step, we select the node with the highest priority, add it to $\mathcal{I}$, and remove the node and its neighbors from $G$. We break ties by selecting the lexicographically smallest node (with 0 smaller than 1). The size of the resulting independent set depends on the choice of the priority function $f$.

In this formulation, constructing large $n$-bit, $s$-deletion-correcting codes reduces to designing a priority function $f$ that maximizes the size of the independent set $\mathcal{I}$. Finding a maximum independent set (MIS) is NP-complete in general (Lovász & Plummer, 2009). We analyze the complexity of our specific problem instances in Appendix C.

## 4 Method

We adapt FunSearch (Romera-Paredes et al., 2024) with a deduplication step to optimize the priority function $f$ to construct large deletion-correcting codes. While conceptually simple, without our deduplication step, we could not find any functions matching conjectured-optimal code sizes. We deduplicate at the output

```
def priority(node, G, n, s):
    # The condition ord(a) > 125 has no effect, as the ASCII values of '0' and '1' are
    always below 125.
    node = ''.join(['-' * (ord(a) > 125) + a for a in list(node)])
    onepositions = [c for c, d in reversed(list(enumerate(node, start=-len(node)))) if d ==
    '1']
    negonesum = sum([-c for c in onepositions])
    # Maximum of negonesum is (n-1)/2 for n odd and n/2 for n even, which is always < n, so
    taking mod n does not change the priority
    finalans = (⌊negonesum/((n + s) · 1)⌋ % n)
    return finalans
```

Figure 3: Discovered priority function provably equivalent to the conjectured-optimal $\mathrm{VT}_0(n)$ code. We added comments for clarity.

level, which requires less compute and is more interpretable than approaches that use embedding models or LLM-as-a-judge (Liu et al., 2025; Lange et al., 2025).

FunSearch works iteratively: in each iteration, we sample existing priority functions as few-shot examples, prompt a pretrained LLM to generate a new one, and evaluate it on a set of search instances (i.e., specific code lengths $n$ and deletion counts $s$). See Figure 1 for an illustration.

### 4.1 Sampling from the database

We organize the priority functions in a database divided into independent sub-databases called islands. We sample few-shot examples from a single island $j$ and store new functions back on the same island, allowing the islands to explore independently.

Within each island, we group priority functions into clusters. Two functions belong to the same cluster $i$ if they achieve the same independent set sizes across all search instances. All functions in a cluster share the same score, which becomes the cluster's $\text{score}_i$.

To sample a priority function, we first sample an island $j$ uniformly at random. We then sample a cluster $i$ from island $j$ with probability

$$p_i = \frac{e^{\text{score}_i / T_j}}{\sum_{i'} e^{\text{score}_{i'} / T_j}}, \quad \text{where } T_j = T\left(1 - \frac{n_j \bmod P}{P}\right).$$

The temperature $T_j$ balances exploration and exploitation on island $j$. It depends on an initial temperature $T$, the number of stored functions $n_j$, and a sampling period $P$. As more functions are stored, the temperature decreases, shifting from uniform sampling (exploration) to favoring high-scoring clusters (exploitation). The temperature resets every $P$ functions to periodically reintroduce exploration. We then sample a function from cluster $i$, favoring shorter ones.

### 4.2 Prompt construction and LLM inference

We construct few-shot prompts by sampling twice from the database to obtain two priority functions. Sampling is done without replacement for diverse examples. The functions are sorted by their cluster scores, with the lower-scoring one first and the higher-scoring one second as target for improvement. The prompt is framed as a code completion task, ending with the header of a new function for the LLM to complete. The initial prompt is shown in Figure 2.

We use StarCoder2-15B (Lozhkov et al., 2024) to generate new priority functions. StarCoder2-15B is an open-access model with 15B parameters trained on The Stack v2 dataset (775B tokens from 600+ programming languages) and additional tokens from sources like pull requests, issues, Jupyter notebooks, and Stack Overflow, totaling 913B tokens.

Table 1: Code sizes for binary single-deletion-correcting codes. Bold indicates best known, t/o no solution found within 48 h (24 h limit, extended to 48 h if no solution was returned).

| | | | | | | | | Code length $n$ | | | | | | | |
|---|---|---|---|---|---|---|---|---|---|---|---|---|---|---|---|
| Method | 6 | 7 | 8 | 9 | 10 | 11 | 12 | 13 | 14 | 15 | 16 | 17 | 18 | 19 | 20 |
| LP upper bound | **10** | 17 | **30** | 53 | 96 | 175 | 321 | 593 | 1104 | 2251 | 4202 | 7882 | 14845 | 28059 | 53202 |
| ILP upper bound | **10** | 16 | **30** | 52 | 94 | 173 | 320 | 592 | 1533 | 2883 | 5441 | t/o | t/o | t/o | t/o |
| ILP lower bound | **10** | **16** | **30** | **52** | **94** | **172** | 242 | 436 | 737 | 1345 | 2468 | 4521 | 8345 | 15495 | 28832 |
| MIS solver: OnlineMIS | **10** | **16** | **30** | **52** | **94** | 143 | 254 | 456 | 816 | 1469 | 2657 | 4843 | 8908 | 16439 | 30440 |
| MIS solver: ReduMIS | **10** | **16** | **30** | **52** | **94** | **172** | **316** | 457 | 812 | 1453 | 2634 | 4803 | 8793 | 16180 | 30111 |
| Random greedy | **10** | **16** | 26 | 42 | 69 | 116 | 201 | 351 | 620 | 1111 | 2003 | 3636 | 6644 | 12215 | 22544 |
| $\mathrm{VT}_0(n)$ code | **10** | **16** | **30** | **52** | **94** | **172** | **316** | **586** | **1096** | **2048** | **3856** | **7286** | **13798** | **26216** | **49940** |
| Ours (search over $s = 1$) | **10** | **16** | **30** | **52** | **94** | **172** | **316** | **586** | **1096** | **2048** | **3856** | **7286** | **13798** | **26216** | **49940** |
| Ours (search over $s \in \{1, 2\}$) | **10** | **16** | **30** | **52** | **94** | **172** | **316** | **586** | **1096** | 2042 | 2474 | 4539 | 8373 | 15451 | 28697 |

### 4.3 Evaluating the new priority function on graphs

We evaluate each new priority function on a set of search instances, each defined by a code length $n$ and deletion count $s$. For each instance, we greedily construct an independent set $\mathcal{I}$ by iteratively adding the highest-priority node to $\mathcal{I}$ and removing it and its neighbors from graph $G$. We pass the graph as an immutable object to the priority function; otherwise, the LLM generates functions that remove edges from $G$ to obtain larger independent sets. This is a form of reward hacking that has also been observed in other works (Georgiev et al., 2025; Zhang et al., 2025).

We discard functions that are not executable (e.g., due to syntax errors). Each executable priority function is assigned a score based on the independent set size it achieves on the search instance with the largest code length $n$, which we found to outperform alternatives such as averaging sizes across instances. After the search, we select the functions achieving the largest independent sets across all search instances and evaluate them on held-out instances with unseen code lengths or deletion counts to test generalization.

### 4.4 Deduplication and storing the new priority function

We store a new priority function on the same island $j$ from which the few-shot examples were sampled. Before storing it, we check for duplicates, i.e., functions that implement the same logic with different syntax. Determining whether two functions are duplicates is non-trivial. We find that comparing at the output level is sufficient: we hash all priority scores a function assigns across all sequences and treat two functions as duplicates if their hashes match. We discard the new function if a matching hash already exists in the cluster; otherwise, we assign it to a cluster based on the independent set sizes it achieves across all search instances, creating a new cluster if no match exists.

### 4.5 Island initialization and resets

Each island is initialized with the same trivial priority function shown in Figure 2, which assigns equal priority to all sequences. To allow information exchange between islands, we periodically reset them. During a reset, we discard the stored priority functions in the worst-performing half of islands, where an island's performance is measured by its highest $\text{score}_i$. We then re-initialize each of these islands with the best-performing function from a random surviving island. We reset islands after every $R$ stored functions rather than after a fixed time interval as in Romera-Paredes et al. (2024), to separate the reset logic from the rate at which functions are generated and evaluated.

## 5 Experiments

We first test whether LLM-guided evolutionary search can match conjectured-optimal code sizes for a single deletion and rediscover the $\mathrm{VT}_0(n)$ code. We also find alternative optimal constructions distinct from any $\mathrm{VT}_a(n)$ code within the search range ($n \leq 11$). However, these do not generalize to longer held-out lengths, and the non-uniqueness of maximum-size single-deletion codes at small lengths was already known.

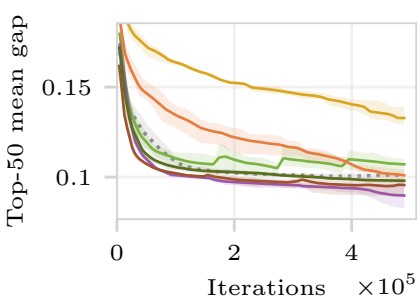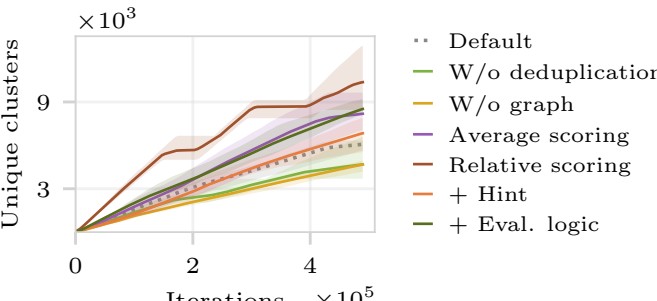

Figure 4: Ablations on single-deletion-correcting codes. Left: gap of the top-50 stored functions (lower is better). Right: unique clusters discovered across all islands (higher is better). Lines are means and shaded bands are min-max ranges across 3 runs.

We then use reasoning models for two- and three-deletion-correcting codes, studying the trade-off between sampling more priority functions and spending more compute per function. We find that, for our problem, sampling more functions is more effective than longer reasoning traces per function.

Motivated by applications in DNA data storage and biological sequencing, we additionally search for quaternary edit codes, finding codes that outperform a recent neural approach (Guo et al., 2025a).

We report results over three runs with fixed random seeds for LLM generation and database sampling, though runs are not fully deterministic due to distributed execution. Our primary metric is the normalized gap to best-known code sizes, averaged over search instances. We report the top-50 mean gap, averaged over the 50 best unique clusters and runs, as a measure of overall search quality. To assess generalization and compare against baselines, we select the function with minimum normalized gap on search instances and evaluate it on held-out instances. We summarize computational costs for each run in Appendix A.

**Baselines.** We compare against explicit constructions from coding theory (Varshamov & Tenengolts, 1965; Helberg & Ferreira, 2002; Abdel-Ghaffar et al., 2011), search-based methods over all binary sequences (Landjev & Haralambiev, 2007), and a greedy construction over $10^5$ random orderings. We additionally compare against general-purpose graph solvers: OnlineMIS and ReduMIS from KaMIS (Dahlum et al., 2016; Lamm et al., 2016) (single-threaded, 3 seeds, 24-hour timeout, best result reported), and Gurobi (16 threads, 24-hour timeout) (Gurobi Optimization, LLC, 2025) as an integer linear programming (ILP) solver. We do not directly compare against neural maximum independent set solvers, as classical solvers currently still outperform them (Böther et al., 2022; Wu et al., 2025). For quaternary edit codes, we additionally compare against DoDo-Code (Guo et al., 2025a), a recent neural approach.

**Upper bounds.** As upper bounds, we solve the linear programming (LP) relaxation of the fractional transversal problem, which gives the tightest bounds for the regimes we consider (Kulkarni & Kiyavash, 2013; Fazeli et al., 2015). Graph statistics for all problem instances are summarized in Appendix G.

### 5.1 Single-deletion correction

We search over code lengths $n \in [6, 11]$, where the maximum code sizes are known and equal to those achieved by the $\text{VT}_0(n)$ code. Smaller code lengths make the problem trivial, while larger lengths result in prohibitive computational and memory costs. Each search runs for 500K iterations, using LLM and evolutionary search hyperparameters from Table 5 in Appendix B, which we found to perform best in smaller-scale experiments.

Table 1 shows that our search matches $\text{VT}_0(n)$ code sizes across all code lengths we consider, while general-purpose graph solvers match the best-known sizes only for small lengths $n < 13$, with the gap increasing with $n$. We find three types of priority functions: (i) functions that construct the same code as $\text{VT}_0(n)$ for all lengths we could verify (including the function in Figure 3, which we prove in Appendix E constructs $\text{VT}_0(n)$ for any length $n$), (ii) functions that construct the $\text{VT}_0(n)$ code at even lengths and its bitwise

Table 2: Code sizes for binary two- and three-deletion-correcting codes. Bold indicates best known, T/O no solution found within 48 h, OOM exceeded 1 TB memory limit.

| Method | Code length $n$ | | | | | | | | | | | | |
|---|---|---|---|---|---|---|---|---|---|---|---|---|---|
| | 6 | 7 | 8 | 9 | 10 | 11 | 12 | 13 | 14 | 15 | 16 | 17 | 18 |
| *2-deletion correction* | | | | | | | | | | | | | |
| LP upper bound | **4** | **5** | **8** | 12 | 20 | 33 | 55 | 93 | 160 | 277 | 487 | 862 | 1539 |
| ILP upper bound | **4** | **5** | **8** | **11** | **16** | **24** | 53 | T/O | T/O | T/O | T/O | T/O | T/O |
| ILP lower bound | **4** | **5** | **8** | **11** | **16** | **24** | 34 | 46 | 72 | 114 | 189 | 316 | 525 |
| MIS solver: OnlineMIS | **4** | **5** | 7 | **11** | **16** | **24** | 36 | 46 | **87** | **136** | **215** | 346 | 563 |
| MIS solver: ReduMIS | **4** | **5** | 7 | **11** | **16** | **24** | 37 | 56 | **87** | 135 | **215** | **349** | **567** |
| Random greedy | **4** | **5** | 7 | 10 | 14 | 20 | 30 | 44 | 66 | 102 | 162 | 258 | 416 |
| Search heuristics (Landjev & Haralambiev, 2007) | **4** | **5** | 7 | **11** | **16** | 21 | 31 | 49 | 75 | 109 | 176 | 286 | 485 |
| Explicit construction (Helberg & Ferreira, 2002) | 3 | 4 | 5 | 6 | 8 | 9 | 11 | 15 | 18 | 22 | 30 | 35 | 43 |
| Ours (search over $s = 2$) | **4** | **5** | 7 | 10 | **16** | 23 | 34 | 48 | 76 | 120 | 194 | 311 | 520 |
| Ours (search over $s \in \{1, 2\}$) | **4** | 4 | 7 | 10 | 15 | 21 | 33 | 49 | 78 | 123 | 200 | 332 | 539 |
| *3-deletion correction* | | | | | | | | | | | | | |
| LP upper bound | **2** | **2** | **4** | **5** | 7 | 10 | 16 | 24 | 38 | 61 | 99 | 163 | 273 |
| ILP upper bound | **2** | **2** | **4** | **5** | **6** | **8** | 16 | T/O | T/O | T/O | T/O | T/O | OOM |
| ILP lower bound | **2** | **2** | **4** | **5** | **6** | **8** | 11 | 13 | 17 | 24 | 36 | OOM | OOM |
| MIS solver: OnlineMIS | **2** | **2** | **4** | **5** | **6** | **8** | **12** | **16** | **22** | **30** | **42** | **60** | T/O |
| MIS solver: ReduMIS | **2** | **2** | **4** | **5** | **6** | **8** | **12** | **16** | T/O | T/O | T/O | T/O | T/O |
| Random greedy | **2** | **2** | **4** | **5** | **6** | **8** | 10 | 13 | 17 | 24 | 32 | 45 | 65 |
| Search heuristics (Landjev & Haralambiev, 2007) | **2** | **2** | **4** | **5** | **6** | 7 | 10 | 12 | 15 | 24 | 31 | 48 | 71 |
| Explicit construction (Helberg & Ferreira, 2002) | **2** | **2** | 3 | 4 | 4 | 5 | 6 | 8 | 8 | 9 | 11 | 15 | 16 |
| Ours (search over $s = 3$) | **2** | **2** | **4** | **5** | **6** | **8** | **12** | **16** | 20 | 25 | 37 | 53 | 78 |
| Ours (search over $s \in \{1, 2\}$) | **2** | **2** | **4** | 4 | 5 | 7 | 10 | 14 | 19 | 28 | 39 | 57 | **81** |

complement $VT_{\frac{n+1}{2}}(n)$ at odd lengths (e.g., Figure 11), and (iii) alternative constructions distinct from any $VT_a(n)$ code within the search range (e.g., Figure 12), though these do not generalize to longer lengths.

While VT codes likely appear in the training data, we argue this is a genuine rediscovery rather than memorization, as the search did not generate VT's standard modular-sum form.

### 5.1.1 Ablations

Our default configuration deduplicates priority functions, includes graph $G$ as input, and scores functions based on the code size achieved at the largest search length $n = 11$. We ablate each design choice below.

**Deduplication.** Without deduplication, none of the three runs find a priority function that achieves optimal code sizes on all search lengths, compared to 2 out of 3 runs with deduplication. Figure 4 (right) shows that without deduplication, the search discovers fewer unique clusters, consistent with 65.8% ($\pm 5.4\%$) of stored functions per island being duplicates.

**Graph input.** Figure 4 (left) shows that the top-50 mean gap is larger without graph $G$ as input to the priority function. Consistently, only 1 out of 3 runs without graph input finds functions achieving optimal code sizes on all search lengths, compared to 2 out of 3 with graph input. Moreover, with graph input, we additionally discover alternative constructions distinct from any $VT_a(n)$ code, such as the function in Figure 12, though these do not generalize to held-out lengths.

**Scoring function.** We consider two alternative scoring functions: (i) averaging code sizes over all lengths $n \in [6, 11]$, and (ii) the average normalized gap to best-known sizes, weighting each length equally. Figure 4 (left) shows that both alternatives achieve a lower top-50 mean gap than the default, which is expected since they directly optimize for average performance. However, our default scoring finds priority functions that achieve optimal code sizes on all search lengths in 2 out of 3 runs, compared to 1 out of 3 for average and 0 out of 3 for normalized gap scoring. This suggests that functions performing well on larger code lengths generalize well to shorter lengths.

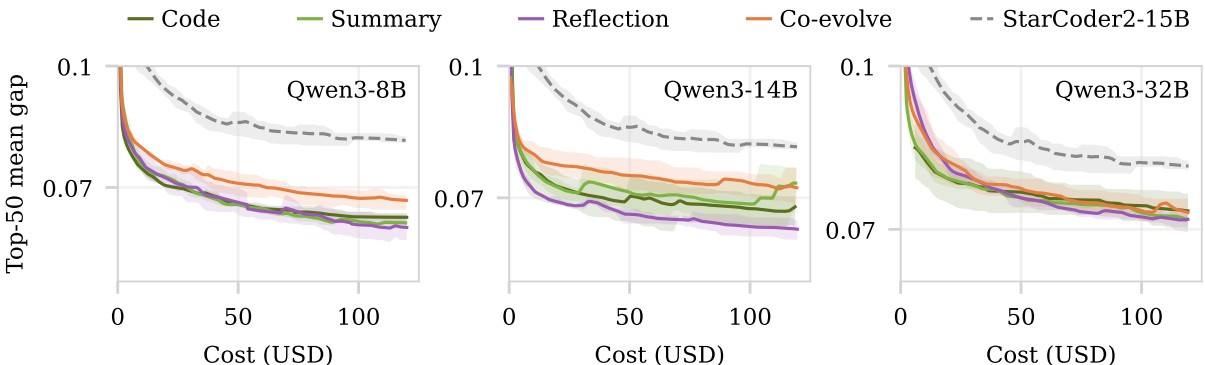

Figure 5: Prompting strategies across Qwen3 model sizes: code (direct solution), summary (algorithmic summary before code), reflection (comparative reflection before code), co-evolve (summary in few-shot examples). Lines are means and shaded bands are min-max ranges across 3 runs.

**Prompt augmentation.** Motivated by position-dependent weighting in VT codes, we add a short instruction to the prompt: *"Consider properties of the binary string 'node', such as specific patterns and the number of ones and zeros."* Under this augmentation, the search discovers the priority function in Figure 3, which is provably equivalent to the $VT_0(n)$ code (see Appendix E for the proof).

With the evaluation logic added as prompt augmentation, i.e., the greedy construction loop that sorts nodes by priority and iteratively builds the independent set, the search does not find a function achieving optimal code sizes on all search lengths.

## 5.2 Two-deletion correction

We search over code lengths $n \in [7, 12]$ with the same setup as for our single-deletion search and augment the docstring of each few-shot example with the per-instance normalized gap to the best-known code size found by general-purpose graph solvers. We evaluate StarCoder2-15B as a baseline and the Qwen3 family (8B, 14B, 32B), which supports toggling reasoning on and off. To test whether our findings hold across model families, we additionally consider the Gemma 4 family (E4B-it, 26B-A4B-it, 31B-it) in Appendix D. Each search runs for a fixed inference budget of \$120, where cost is defined as $C = T_{in} p_{in} + T_{out} p_{out}$, with $T_{in}$ and $T_{out}$ the input and output token counts and $p_{in}, p_{out}$ the per-token prices.

Table 2 compares the code sizes constructed by our best priority function (Figure 14) and baselines. Our function outperforms the explicit construction of Helberg & Ferreira (2002) across all lengths we consider. Compared to search-based methods in sequence space (Landjev & Haralambiev, 2007), it achieves competitive or larger code sizes at longer held-out lengths, despite not being directly optimized for them. This suggests that searching in function space generalizes better than searching in sequence space, which becomes exponentially harder with code length. Across all discovered functions, we match the code sizes of general-purpose graph solvers at search lengths $n \in [7, 12]$, though our best function constructs smaller codes at $n = 9, 11$ within the search range and at all held-out lengths. We analyze this function in Appendix F.

### 5.2.1 Ablations

We study two axes of investing more compute per function, prompting strategy and reasoning mode. For prompting strategy, we vary how the model generates the priority function: (a) code only, (b) an algorithmic summary before code, (c) a comparative reflection before code (Ye et al., 2024), or (d) code and algorithmic summary co-evolved jointly (Liu et al., 2024). For reasoning mode, we toggle it on and off across all prompting strategies. We increase maximum new tokens from 246 (used for StarCoder2-15B) to 2048 with reasoning off and 16K with reasoning on. We also increase the sampling period $P$ from 30K to 180K and the number $R$ of functions stored before island reset from 1.2K to 6K to account for the lower execution failure rate of Qwen3 and Gemma 4 compared to StarCoder2-15B. For Qwen3-8B and Qwen3-14B we use the same temperature, top-$p$, and repetition penalty as for StarCoder2-15B, which we found to perform similarly to

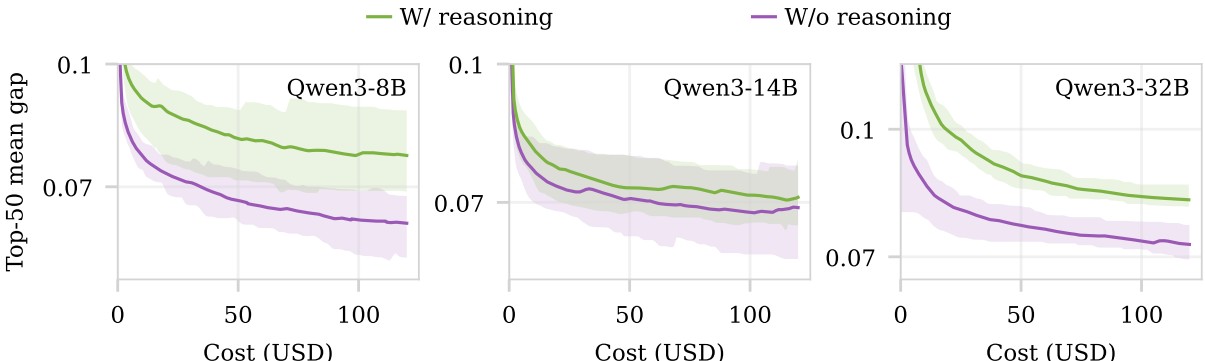

Figure 6: Reasoning mode across Qwen3 model sizes, averaged over prompting strategies. Lines are means and shaded bands are min-max ranges across 12 runs.

or slightly better than each model's recommended defaults (see Figure 8 in Appendix B.1). For Qwen3-32B and the Gemma 4 family we use each model's recommended defaults.

**Co-evolving natural language with code hurts search quality.** Figure 5 shows that across all Qwen3 model sizes, co-evolving an algorithmic summary with code results in a worse top-50 mean gap. We observe that co-evolved descriptions become increasingly abstract over time, for example *"Fuse multi-scale structural analysis with reinforcement learning-inspired adaptivity to dynamically balance immediate gains against future expansion potential during greedy selection."* This prompts the model to attempt ideas it cannot correctly implement, resulting in higher execution failure rates of 71%, 76%, and 40% for the 8B, 14B, and 32B models, respectively, compared to 14%, 50%, and 22% for code only. In Appendix D, we show that this finding also holds for the Gemma 4 family.

**Sampling more functions is more effective than longer reasoning traces.** Figure 6 shows that for the Qwen3 family, toggling reasoning on does not improve solution quality and, under a fixed inference budget, has a worse top-50 mean gap as fewer functions can be evaluated. Runs with reasoning enabled evaluate on average 2.1×, 1.4×, and 8.7× fewer priority functions for Qwen3-8B, 14B, and 32B, respectively. Interestingly, even when controlling for the number of priority functions evaluated, reasoning performs similarly to or worse than non-reasoning. By inspection, we find that while reasoning traces explore complex and novel ideas, the model frequently concludes that more complex approaches are too computationally expensive and that novel ideas lack sufficient justification, converging instead to generic low-degree heuristics. As a proxy for algorithm simplicity, we compute average lines of code, percentage of generated functions with helper functions, and cyclomatic complexity (number of conditionals such as `if` and `while`), finding that runs with reasoning toggled on consistently generate priority functions with fewer lines of code (26.9 vs. 35.8), fewer helper functions (20.1% vs. 34.4%), and lower cyclomatic complexity (7.6 vs. 9.9) across all models and prompting strategies. In Appendix D, we show that toggling reasoning on also gives a higher top-50 mean gap for the Gemma 4 family, suggesting that for our problem, sampling more priority functions is more effective than longer reasoning traces per function.

### 5.3 Generalization across deletion counts

We now study whether a function optimized jointly for single- and two-deletion correction generalizes to three-deletion correction. We search over code lengths $n \in [9, 11]$ for $s = 1$ and $n \in [10, 12]$ for $s = 2$, and evaluate the best function on three-deletion-correcting codes of lengths $n \in [6, 18]$, comparing against a function directly optimized for $s = 3$ on lengths $n \in [8, 13]$.

Table 2 shows that on search lengths $n \in [8, 13]$, the function in Figure 16 jointly optimized for $s = 1, 2$ has a gap of 14.3% to best-known code sizes, versus zero gap for the function in Figure 15 directly optimized for $s = 3$. However, on held-out lengths, the joint function achieves a lower gap of 5.41% compared to 8.50% for the function optimized for $s = 3$. We observe a similar trend for two deletions, where the joint function constructs strictly larger codes for all held-out lengths $n > 12$. This suggests that jointly optimizing

Table 3: Code sizes for quaternary single-edit-correcting codes. Bold indicates best known, T/O no solution found within 48 h, OOM exceeded 1 TB memory limit, "–" results not reported in the original paper.

| | Code length $n$ | | | | | |
|---|---|---|---|---|---|---|
| Method | 6 | 7 | 8 | 9 | 10 | 11 |
| LP upper bound | 215 | 744 | 2621 | 9362 | 33825 | 123361 |
| ILP upper bound | 212 | 853 | 4527 | 131072 | T/O | OOM |
| ILP lower bound | 106 | 311 | 1027 | 3457 | 11743 | OOM |
| MIS solver: OnlineMIS | 112 | 345 | 1119 | 3678 | 12399 | **42080** |
| MIS solver: ReduMIS | **113** | **352** | **1130** | **3717** | **12489** | T/O |
| Random greedy | 91 | 269 | 844 | 2747 | 9182 | 31554 |
| DoDo-Code (Guo et al., 2025a) | – | 275 | 900 | 3011 | 10414 | 36368 |
| Ours (search over $s = 1$) | 101 | 321 | 1038 | 3443 | 11743 | 40604 |

across deletion counts acts as a regularizer, helping discover functions that generalize better to unseen code lengths, consistent with Georgiev et al. (2025), who find that searching jointly over related problem instances improves generalization.

For $s = 1$, Table 1 shows that the joint function matches optimal code sizes for $n \leq 14$, outperforming general-purpose graph solvers, but constructs smaller codes than the function directly optimized for $s = 1$ at longer lengths. Interestingly, the searches for two-, three-, and joint deletion correction all converge to graph-based priority functions, while the search for single-deletion correction converges to functions that use only algebraic properties of the binary string.

### 5.4 Extension to single-edit codes and non-binary alphabets

While our approach currently does not scale well to moderate or long code lengths, short codes are practically relevant in biological sequencing and DNA data storage. In biological sequencing, short barcodes are used for sample identification and need to be distinguishable under insertions, deletions, and substitutions (edit errors) (Craig et al., 2008). In DNA data storage, current synthesis technologies do not support long DNA strands and information must be broken into short pieces that are stored in unordered pools. These sequences require indices that are similarly distinguishable under edit errors to allow correct reordering (Antkowiak et al., 2020) and random access (Organick et al., 2018). Therefore, we additionally search for quaternary single-edit-correcting codes using the best-performing configuration from our two-deletion experiments.

Table 3 shows the code sizes achieved by our priority function in Figure 13 and baselines for quaternary single-edit-correcting codes. Our function outperforms DoDo-Code (Guo et al., 2025a) across all lengths considered. DoDo-Code uses the same greedy construction as our approach, but with a fixed, hand-designed heuristic that adds sequences with the smallest edit ball first, where edit balls are approximated via learned edit distance embeddings. Compared to general-purpose graph solvers, our function constructs codes similar in size to those from the ILP solver, but smaller than those from the KaMIS solvers.

## 6 Scalability

A key limitation of our approach is the scalability of the evaluator, which makes using our priority functions at larger code lengths infeasible. Our greedy construction algorithm requires computing priorities for all sequences and therefore scales exponentially in code length.

To use our priority functions at larger code lengths, we need an efficient encoder that maps a message to a codeword without enumerating all sequences. Constructing such an encoder from a discovered priority function requires (1) characterizing which priority scores belong to the code for any length $n$, and (2) determining how to insert redundant bits to achieve a target priority score. We demonstrate (1) for the function in Figure 3 (Appendix E), where (2) follows from the known $\text{VT}_0(n)$ encoder. Deriving similar characterizations or encoders for our multiple-deletion functions remains an open problem.

Nevertheless, compared to previous search-based methods (Landjev & Haralambiev, 2007; Swart & Ferreira, 2003), the discovered priority functions can be analyzed mathematically, which is harder for codes found by direct search. An advantage over general-purpose graph solvers is that evaluating our functions does not require the full confusability graph, which quickly becomes memory-infeasible. As long as the priority function does not depend on global graph statistics (which is the case for our best discovered functions), priority computation per sequence is polynomial in $n$.

## 7 Conclusion

In this work, we showed that LLM-guided evolutionary search is a promising tool for constructing error-correcting codes, rediscovering conjectured-optimal codes for a single binary deletion. For multiple binary deletions and quaternary edit correction, we found larger codes at short lengths than prior search-based and explicit constructions, though these codes remain primarily empirical, with limited theoretical understanding. We studied key design choices and found that, for our problem, sampling more functions is more effective than investing more compute per function, and that deduplication is critical for search diversity.

## Acknowledgments

Funded by the European Union (DiDAX, 101115134). Views and opinions expressed are however those of the author(s) only and do not necessarily reflect those of the European Union or the European Research Council Executive Agency. Neither the European Union nor the granting authority can be held responsible for them.

The authors thank Maria Abu Sini for proofreading and helpful comments, as well as Roni Con and Eitan Yaakobi for insightful discussions on deletion-correcting codes.

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

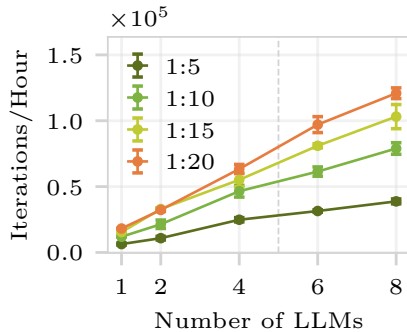

Figure 7: Iterations per hour for different LLM-to-evaluator ratios in our distributed implementation of FunSearch. Ratios to the right of the dashed line use two nodes.

Table 4: Average computational cost per run (number of runs in parentheses) where -R denotes reasoning enabled.

| Model | GPUs | Cores | Wall time (h) | GPU hours | CPU hours | Throughput (tok/s) |
|---|---|---|---|---|---|---|
| StarCoder2-15B (10) | 4×A100-80GB | 96 | 7.84 ± 0.28 | 31.16 ± 1.15 | 39.68 ± 13.77 | 3957.3 ± 117.2 |
| Qwen3-8B (2) | 4×H100-94GB | 96 | 19.91 ± 7.04 | 44.18 ± 2.10 | 1221.33 ± 620.77 | 3795.8 ± 1373.7 |
| Qwen3-14B (10) | 4×H100-94GB | 96 | 22.99 ± 5.59 | 57.42 ± 5.35 | 798.83 ± 224.08 | 3514.6 ± 1089.6 |
| Qwen3-32B (10) | 4×H100-94GB | 96 | 9.42 ± 3.11 | 35.2 ± 14.8 | 245.1 ± 65.5 | 2046.1 ± 827 |
| Qwen3-8B-R (2) | 4×A100-40GB | 128 | 25.41 ± 1.82 | 101.26 ± 7.09 | 632.22 ± 146.56 | 4534.5 ± 292.9 |
| Qwen3-14B-R (10) | 4×H100-94GB | 96 | 25.65 ± 3.80 | 95.77 ± 6.42 | 613.15 ± 200.88 | 4330.9 ± 944.2 |
| Qwen3-32B-R (5) | 4×H100-94GB | 96 | 25.51 ± 11.6 | 100.5 ± 46.2 | 15.6 ± 3.7 | 1533.5 ± 601 |

# A    Implementation details

We implement FunSearch using RabbitMQ for parallelization via asynchronous message passing. The implementation consists of multiple LLMs and evaluators, and a single program database, all running as independent workers communicating through message queues. The database constructs prompts and sends them to the LLM queue, the LLMs generate new priority functions and forward them to the evaluator queue, and the evaluators compute scores and return results to the database. Table 4 summarizes the computational cost per run for each model used in our experiments.

Our implementation differs from existing open-source implementations such as OpenEvolve (Sharma, 2025), EoH (Liu et al., 2024), ReEvo (Ye et al., 2024), and CodeEvolve (Assumpção et al., 2025) in two ways. First, existing implementations parallelize by iteration, running multiple independent copies of the full loop (generate, evaluate, and store) as separate processes on a single node, where each copy blocks until the current step completes. In contrast, we parallelize by task using asynchronous message passing, so no worker has to wait for another (e.g., evaluators process previously generated functions while the LLM generates new ones) and workers can be distributed across multiple nodes. Second, existing implementations rely on API-based inference, where requests are processed individually, while we use batched LLM inference with locally hosted models.

We measure throughput in iterations per hour, which depends on the number of LLMs and evaluators. We allocate one GPU (NVIDIA A100, 80GB) per LLM, one CPU core per evaluator, and one additional core for the database, with a 30-second evaluator timeout. Throughput is measured on the single-deletion-correcting code search ($s = 1$, $n \in [6, 11]$) using StarCoder2-15B. Figure 7 shows throughput at different LLM-to-evaluator ratios, measured over four 20-minute windows (mean ± SD). For each ratio, we report results at the optimal configuration found via grid search over LLM batch size (25, 50, 100, 150), evaluator prefetch count (5, 15, 30), and parallel workers per evaluator (1, 2, 3). The LLM prefetch count is fixed at twice the

Table 5: LLM and evolutionary search hyperparameters used in our main experiments. Qwen3-8B and Qwen3-14B reuse StarCoder2-15B's sampling hyperparameters. Qwen3-32B and Gemma 4 use each model's recommended defaults. Reasoning-on values are shown in parentheses where they differ and Top-$k = -1$ denotes no top-$k$ filtering.

| Model | Max. new tokens | Repetition penalty | Temperature | Top-$p$ | Top-$k$ | $T$ | $P$ | $R$ |
|---|---|---|---|---|---|---|---|---|
| StarCoder2-15B | 246 | 1.22 | 0.9444 | 0.7778 | -1 | 0.1 | 30K | 1.2K |
| Qwen3-8B/14B | 2048 (16K) | 1.22 | 0.9444 | 0.7778 | -1 | 0.1 | 180K | 6K |
| Qwen3-32B | 2048 (16K) | 1.0 | 0.7 (0.6) | 0.8 (0.95) | 20 | 0.1 | 180K | 6K |
| Gemma 4 | 2048 (16K) | 1.0 | 1.0 | 0.95 | 64 | 0.1 | 180K | 6K |

batch size, allowing the next batch to buffer during GPU inference, which dominates processing time. We scale from 1 to 8 LLMs across one or two nodes, scaling evaluators proportionally.

Throughput scales near-linearly with the number of LLMs, showing that message passing overhead is negligible compared to the cost of LLM inference and evaluation. At a ratio of 1:20, for example, throughput increases from approximately 18K iterations per hour with 1 GPU to 120K with 8 GPUs across two nodes, with no significant overhead from scaling to multiple nodes. We achieve the highest throughput at the largest tested ratio of 20 evaluators per LLM and expect further gains from additional evaluators up to a point. However, we note that the optimal ratio is specific to our setup, as it depends on the hardware, the LLM used, the evaluator timeout, and the problem instances evaluated (e.g., harder instances take longer to evaluate relative to LLM inference time). Our implementation also supports dynamically scaling the number of LLMs and evaluators during execution based on message load and available resources.

# B  Hyperparameter optimization

We optimize hyperparameters for both the LLM and the evolutionary search. For the LLM, we consider maximum new tokens, repetition penalty, temperature, top-$p$, and top-$k$. For the evolutionary search, we consider initial temperature $T$, sampling period $P$, and the number of functions $R$ stored before an island reset. Table 5 summarizes the hyperparameter values used for each model in the main paper.

## B.1  LLM hyperparameters

For StarCoder2-15B, we tune LLM hyperparameters via grid search on the single-deletion-correcting code search ($s = 1$, $n \in [6, 11]$). We run two independent grid searches, varying maximum new tokens and repetition penalty while keeping temperature and top-$p$ fixed, and vice versa. We measure the average improvement in independent set sizes constructed by the best priority functions across all islands, relative to the trivial initialization. Each run is evaluated after one hour using one GPU and 40 CPUs.

For maximum new tokens, we consider values in $[60, 300]$, and for repetition penalty, values in $[1.0, 2.0]$, both divided into 10 equally spaced grid points. Temperature and top-$p$ are fixed at 0.2 and 0.95, respectively, as in Section 7.1.3 of Lozhkov et al. (2024). Low repetition penalties combined with high maximum new tokens often result in the LLM repeating the code completion task, generating multiple function headers with minor variations or trivial return statements instead of a single, improved function. Repetition penalties above 1.22 fail to generate executable functions. While we achieve competitive results with maximum new tokens between 60 and 140 and repetition penalties between 1.05 and 1.11, we observe the largest improvement with maximum new tokens 246 and a repetition penalty of 1.22. Since we only need to find one good priority function to discover a new maximum code size, we use 246 for maximum new tokens and 1.22 for the repetition penalty.

For temperature and top-$p$, we consider values in $[0.5, 1.5]$ and $[0.6, 1.0]$, respectively, with 10 equally spaced grid points, while keeping maximum new tokens at 246 and repetition penalty at 1.22. Higher variability in token sampling (larger temperature and top-$p$ values) leads to more variable code sizes across runs, but also to larger best-found sizes. More deterministic sampling generates more syntactically correct functions but does not lead to larger code sizes. These findings align with the hypothesis of Romera-Paredes et al. (2024)

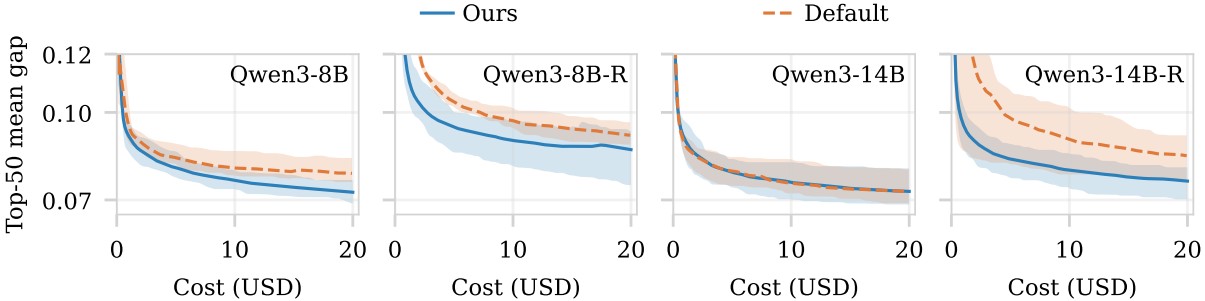

Figure 8: Top-50 mean gap on small-scale two-deletion experiments ($20 budget) comparing our hyperparameters (Ours) against each model's recommended defaults (Default), without and with (-R) reasoning. Lines are means and shaded bands are min-max ranges across prompting strategies.

that the LLM contributes by exploring diverse functions, occasionally generating good executable ones but often producing unusable outputs. We achieve the best results at a temperature of 0.9444 and a top-$p$ of 0.7778.

For the Qwen3 and Gemma 4 model families, we increase maximum new tokens from 246 to 2048 after observing many truncated functions at 246 tokens. With reasoning toggled on, we further increase maximum new tokens to 16K to allow for longer reasoning traces. For temperature, top-$p$, and repetition penalty, we use each model's recommended defaults from its model card, except for Qwen3-8B and Qwen3-14B. For these two models, we reuse the values tuned for StarCoder2-15B, since Figure 8 shows that they perform similarly or slightly better than the recommended defaults across both reasoning modes.

### B.2 Evolutionary search hyperparameters

For StarCoder2-15B, we tune all three evolutionary search hyperparameters via grid search on the single-deletion-correcting code search ($s = 1$, $n \in [6, 11]$), using the best-performing LLM hyperparameters from Appendix B.1. We measure performance as a binary outcome: whether or not we find a priority function that constructs optimal code sizes on all search instances. Each run runs for 400K iterations or stops early if such a function is found.

For initial temperatures $T \in \{0.05, 0.1, 0.3, 0.5, 1\}$ with fixed sampling period $P = 30K$ and $R = 1.2K$ functions stored before a reset, we find a priority function that constructs optimal code sizes on all search instances only when $T = 0.1$. With $T = 0.1$, we find such a function after $\sim$116K iterations, with 20.7% of generated functions stored at the end of the search. When $T = 0.05, 0.3, 0.5$, or 1, the percentages of stored functions are 18.6%, 19.3%, 12.0%, and 10.0%, respectively.

For sampling periods $P \in \{5K, 10K, 30K, 50K, 100K\}$, with $T = 0.1$ and $R = 1.2K$, adjusting the sampling period does not improve performance beyond $P = 30K$. With $P = 5K$, we find a priority function that constructs optimal code sizes on all search instances after $\sim$194K iterations, with 18.1% stored at the end of the search. With $P = 50K$, we find such a function after $\sim$132K iterations, with 23.0% stored. When $P = 10K$ or $P = 100K$, we do not find such a function after 400K iterations, with 13.0% and 19.8% stored, respectively.

For $R \in \{300, 600, 1.2K, 2.4K, 5K\}$, with $T = 0.1$ and $P = 30K$, varying $R$ does not improve performance beyond $R = 1.2K$. With $R = 300$, we find a priority function that constructs optimal code sizes on all search instances after $\sim$251K iterations, with 18.2% stored at the end of the search. With $R = 600$, we find such a function after $\sim$197K iterations, with 19.9% stored. When $R = 2.4K$ or $R = 5K$, we do not find such a function within 400K iterations, with 19.2% and 19.6% stored, respectively.

We also experiment with dynamically decreasing the LLM sampling temperature to balance exploration and exploitation during the search. The temperature starts at 0.94 and decreases as more functions are

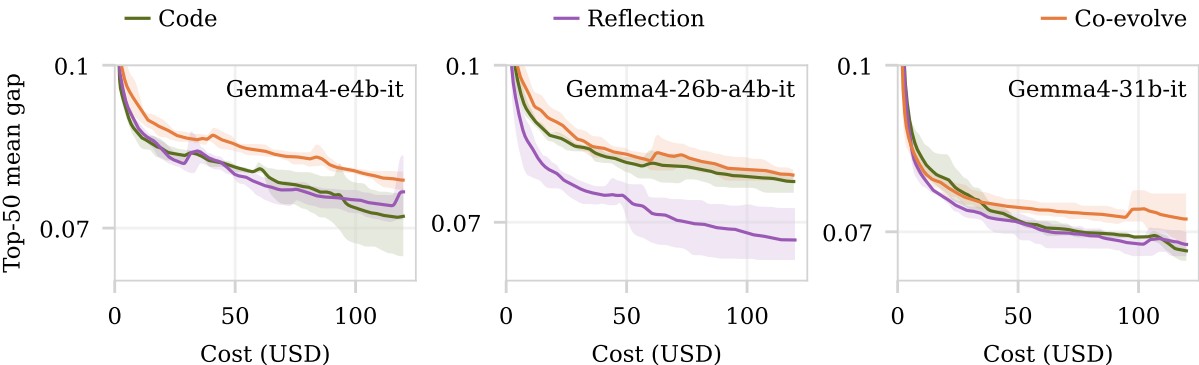

Figure 9: Prompting strategies across Gemma 4 model sizes: code (direct solution), reflection (comparative reflection before code), co-evolve (summary in few-shot examples). Lines are means and shaded bands are min-max ranges across 3 runs.

stored on the island from which the prompt is sampled, reaching zero at $D \in \{5K, 10K, 20K, 50K\}$ stored functions. While this approach slightly increases the number of executable functions, it does not improve search efficiency compared to a fixed temperature. With $D = 5K$, we find a priority function that constructs optimal code sizes on all search instances after $\sim$247K iterations, with 22.6% stored at termination. When $D = 10K$, $20K$, or $50K$, we do not find such a function within 400K iterations, with 21.1%, 17.2%, and 21.4% stored, respectively.

For the Qwen3 and Gemma 4 model families, we increase the sampling period from $P = 30$K to $P = 180$K and the number of functions stored before island reset from $R = 1.2$K to $R = 6$K, because the execution failure rate is lower. On Qwen3-8B, we observe that keeping $P$ and $R$ at their StarCoder2-15B values causes island resets to happen too frequently, resulting in a noisier top-50 mean gap curve. We keep the initial temperature at $T = 0.1$, as it does not depend on the number of stored functions.

## C  Problem complexity

Finding a maximum independent set in a general graph is NP-complete (Lovász & Plummer, 2009). If we know the maximum code size $C^*$, we must evaluate all $\binom{2^n}{C^*}$ subsets of that size. For code length $n = 6$ (the smallest length we consider), deletion count $s = 1$, and $C^* = 10$, this already exceeds 151 billion subsets. If we do not know the maximum code size, we must consider all possible subset sizes, giving $2^{2^n}$ subsets in total. Verifying whether a single subset of size $k$ forms a valid code requires checking that no two sequences share a common subsequence of length $n - s$, which takes $O(n^2)$ time per pair using dynamic programming (Cormen et al., 2022), and $O(k^2 n^2)$ time in total. This gives a worst-case complexity of $O(2^{2^n} \cdot k^2 n^2)$ for exhaustive search over all subsets.

## D  Additional results

In this section, we additionally evaluate the prompting strategy and reasoning mode ablations from Section 5.2.1 on the Gemma 4 family. Since LiteLLM does not yet provide pricing for Gemma 4 models, we approximate cost using Fireworks AI entries for similarly sized Qwen3 models, matching by parameter count and architecture (dense vs. MoE). Specifically, we use Qwen3-4B for Gemma 4-E4B-it (4B dense), Qwen3-30B-A3B for Gemma 4-26B-A4B-it (MoE), and Qwen3-32B for Gemma 4-31B-it (dense).

Figure 9 shows that, consistent with the Qwen3 family, co-evolving an algorithmic summary with code hurts overall search quality for all three Gemma 4 model sizes. Figure 10 shows that toggling reasoning on also results in a higher top-50 mean gap, consistent with our finding that for our problem, sampling more priority functions is more effective than longer reasoning traces per function.

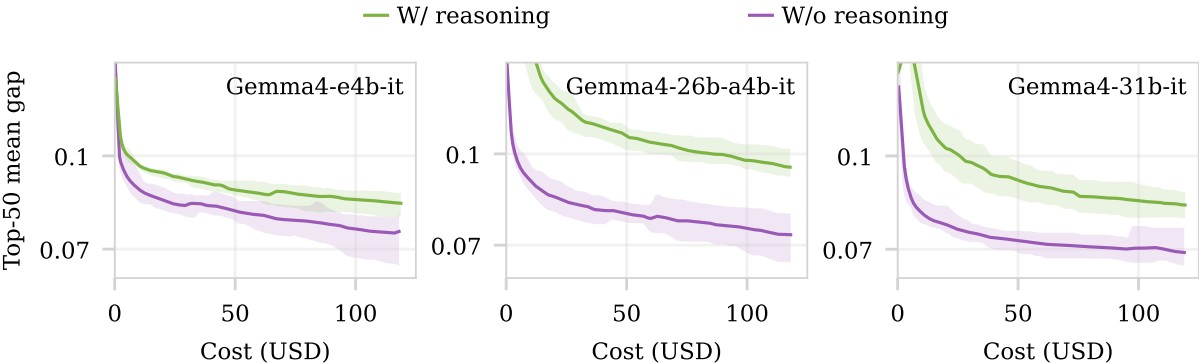

Figure 10: Reasoning mode across Gemma 4 model sizes, averaged over prompting strategies. Lines are means and shaded bands are min-max ranges across 9 runs.

## E  Equivalence to VT codes

In this section, we prove that our discovered priority function $f$ in Figure 3 constructs the largest $\text{VT}_0(n)$ code for any code length $n$ under greedy construction with lexicographic tie-breaking.

For a single deletion, the function assigns priority to a binary sequence $\mathbf{x} \in \{0,1\}^n$ as

$$f(\mathbf{x}) = \left\lfloor \frac{W(\mathbf{x})}{n+1} \right\rfloor \quad \text{where} \quad W(\mathbf{x}) = \sum_{i=1}^{n}(n-i+1)x_i. \tag{1}$$

Since $(n-i+1) \equiv -i \pmod{n+1}$, we have $\mathbf{x} \in \text{VT}_a(n)$ with $a \equiv -W(\mathbf{x}) \pmod{n+1}$ and it suffices to show that any sequence $\mathbf{w} \notin \text{VT}_0(n)$ conflicts with some sequence $\mathbf{v} \in \text{VT}_0(n)$ that the greedy algorithm selects first.

By the maximality of VT codes (Cullina et al., 2012), there exists a sequence $\mathbf{v} \in \text{VT}_0(n)$ that shares a length-$(n-1)$ subsequence $\mathbf{z}$ with $\mathbf{w}$. Write $\mathbf{v} = I_{b_v}^{j_v}(\mathbf{z})$ and $\mathbf{w} = I_{b_w}^{j_w}(\mathbf{z})$, where $b_v, b_w \in \{0,1\}$ and $j_v, j_w$ are the bits and positions inserted into $\mathbf{z}$ to obtain sequences $\mathbf{v}$ and $\mathbf{w}$, respectively.

**Claim 1.** $f(\mathbf{v}) \in \{f(\mathbf{w}), f(\mathbf{w})+1\}$.

*Proof of Claim 1.* Let $q = f(\mathbf{w})$ and $r = W(\mathbf{w}) \bmod (n+1)$, so that $W(\mathbf{v}) = (n+1)f(\mathbf{v})$ and $W(\mathbf{w}) = (n+1)q + r$ with $r \in \{1,\dots,n\}$. Then

$$W(\mathbf{v}) - W(\mathbf{w}) = (n+1)(f(\mathbf{v}) - q) - r.$$

We bound $|W(\mathbf{v}) - W(\mathbf{w})|$ by case analysis on $(b_v, b_w)$. Let $m \leq n-1$ denote the number of 1-bits in $\mathbf{z}$.

**Case A:** $b_v = b_w = 0$. Inserting a 0-bit at different positions shifts at most $m$ ones by one position, giving $|W(\mathbf{v}) - W(\mathbf{w})| \leq m \leq n-1$.

**Case B:** $b_v = b_w = 1$. Inserting a 1-bit at position $j$ adds $n-j+1$ to $W(\mathbf{z})$, ranging from $m+1$ to $n$, giving $|W(\mathbf{v}) - W(\mathbf{w})| \leq n - (m+1) \leq n-1$.

**Case C:** $b_v \neq b_w$. Without loss of generality, assume $b_v = 1$ and $b_w = 0$. Then $W(\mathbf{v}) = W(\mathbf{z}) + (n-j_v+1)$ and $W(\mathbf{z}) - m \leq W(\mathbf{w}) \leq W(\mathbf{z})$, giving $1 \leq W(\mathbf{v}) - W(\mathbf{w}) \leq (n-j_v+1) + m \leq n$.

In all cases, $|W(\mathbf{v}) - W(\mathbf{w})| \leq n$, so $|(n+1)(f(\mathbf{v}) - q) - r| \leq n$. Since $r \in \{1,\dots,n\}$, this gives $f(\mathbf{v}) - q \in \{0,1\}$. □

**Claim 2.** If $f(\mathbf{v}) = f(\mathbf{w})$, then $\mathbf{v} <_{\text{lex}} \mathbf{w}$.

```python
def priority(node, n, s):
    count = 0
    for i in range(len(node)):
        for j in range(i + 1, len(node) + 1):
            substr = ''.join(node[i:j])
            count += substr.count("0") ** 3
    return -(6 * count / 2 ** (n + 1) - n ** 2)
```

Figure 11: Priority function equal to the $\text{VT}_0(n)$ code at even lengths and its complement $\text{VT}_{\frac{n+1}{2}}(n)$ at odd lengths, verified for $n \in [6, 25]$. The priority can be simplified to $\sum_{1 \leq i \leq j \leq n} \left( \sum_{k=i}^{j} \mathbb{1}[x_k = 0] \right)^2$ and constructs the same codebook under arbitrary tie-breaking.

*Proof of Claim 2.* If $f(\mathbf{v}) = f(\mathbf{w})$, then $W(\mathbf{w}) = W(\mathbf{v}) + r > W(\mathbf{v})$. We show that $\mathbf{w} <_{\text{lex}} \mathbf{v}$ implies $W(\mathbf{w}) \leq W(\mathbf{v})$, a contradiction, so we must have $\mathbf{v} <_{\text{lex}} \mathbf{w}$.

**Case A:** $b_v = b_w = 0$ with $j_w < j_v$ (so $\mathbf{w} <_{\text{lex}} \mathbf{v}$). For positions $i \in [j_w, j_v)$, bits contribute weight $(n - i)$ in $\mathbf{w}$ versus $(n - i + 1)$ in $\mathbf{v}$, giving $W(\mathbf{v}) - W(\mathbf{w}) = \sum_{i=j_w}^{j_v - 1} z_i \geq 0$.

**Case B:** $b_v = b_w = 1$ with $j_w > j_v$ (so $\mathbf{w} <_{\text{lex}} \mathbf{v}$). The inserted 1-bit contributes $(n - j_v + 1)$ to $W(\mathbf{v})$ and $(n - j_w + 1)$ to $W(\mathbf{w})$. Since $j_w > j_v$, we have $W(\mathbf{v}) > W(\mathbf{w})$.

**Case C:** If $b_v = 1$ and $b_w = 0$, then $W(\mathbf{v}) \geq W(\mathbf{w}) + 1$ from Case C of Claim 1. If $b_v = 0$ and $b_w = 1$, then $\mathbf{w}$ cannot be lexicographically smaller than $\mathbf{v}$.

In all cases, $\mathbf{w} <_{\text{lex}} \mathbf{v}$ implies $W(\mathbf{w}) \leq W(\mathbf{v})$, contradicting $W(\mathbf{w}) > W(\mathbf{v})$. $\qquad\square$

By Claims 1 and 2, $\mathbf{v}$ either has strictly higher priority or is lexicographically smaller at the same priority, so the greedy algorithm selects sequence $\mathbf{v}$ before $\mathbf{w}$ for all $\mathbf{w} \notin \text{VT}_0(n)$, completing the proof.

# F  Discovered priority functions

In this section, we analyze our best function for two-deletion correction (Figure 14) as a representative example.

The function computes the priority as a product of six heuristics with different exponents. To analyze the contribution of each heuristic, we ablate each individually by setting its exponent to zero and measuring the mean relative decrease in code size over lengths $n \in [6, 12]$ and deletion count $s = 2$. We observe the largest average decrease in code size of 30% when removing the heuristic that assigns higher priority to low-degree nodes, a common greedy heuristic for maximum independent set problems. The second largest average decrease of 15% occurs when removing the heuristic that assigns higher priority to sequences with a smaller deletion ball, i.e., fewer distinct subsequences obtained by deleting up to $s$ bits. This is consistent with prior work (Guo et al., 2025a), which also uses the edit ball size as a heuristic for constructing large edit codes. Removing the heuristic that assigns higher priority to sequences with low binary entropy, i.e., an unbalanced number of zeros and ones, decreases code sizes on average by 6.3%. This is consistent with prior work showing that uniform input distributions do not achieve the capacity of the deletion channel (Drmota et al., 2012).

The remaining heuristics contribute less to code sizes. Removing the heuristic that assigns higher priority to sequences whose average Hamming distance to their neighbors is close to $n/2$ decreases code sizes on average by 3.7%, removing the heuristic that assigns higher priority to sequences with lower variance in Hamming distance to their neighbors decreases code sizes on average by 1.9%, and removing the heuristic that assigns higher priority to sequences whose neighbors have high average degree decreases code sizes on average by 0.6%.

The other functions that achieve the same minimum normalized gap on search instances $n \in [7, 12]$ follow similar logic, using different weightings of the heuristics, a subset of them, or a weighted sum rather than a product.

```
def priority(node, G, n, s):
    ones = sum(int(c) for c in node)
    max_run = 0
    run = 0
    for c in node:
        if c == '1':
            run += 1
            max_run = max(max_run, run)
        else:
            run = 0
    is_sorted = (max_run == ones) if ones > 0 else True

    scores = []
    for nb in G.neighbors(node):
        score = sum(-ord(nb[-i]) * pow(i, (n - (i + 1)))
                    * (1 + 0.2 * min(i, n + 1 - i))
                    for i in range(1, len(nb) + 1))
        scores.append(score)
    nb_max = max(scores) if scores else 0

    eff_ones = n // 2 if ones == n else ones
    base = nb_max * eff_ones * eff_ones
    return base + 1e-12 * (1 if is_sorted else 0)
```

Figure 12: Priority function that constructs single-deletion-correcting codes matching best-known sizes for $n \leq 12$, with codes distinct from all $\text{VT}_a(n)$ codes at lengths $n = 6, 7, 9, 10, 11$.

```
def priority(node, G, n, s):
    chars = [ord(c) for c in node]

    degree_penalty = -len(list(G.neighbors(node)))

    unique_chars, counts = np.unique(chars, return_counts=True)
    probabilities = counts / n
    shannon_entropy = -np.sum(probabilities * np.log2(probabilities + 1e-10))

    char_positions = {}
    for index, char in enumerate(chars):
        if char not in char_positions:
            char_positions[char] = []
        char_positions[char].append(index)

    distribution_spread = 0
    for positions in char_positions.values():
        mean_pos = np.mean(positions)
        dist_from_mean = [(pos - mean_pos)**2 for pos in positions]
        distribution_spread += np.sqrt(np.mean(dist_from_mean))

    lex_order_weight = -float(int(''.join(map(str, chars))))

    final_priority = (
        0.4 * degree_penalty +
        0.35 * shannon_entropy +
        0.15 * distribution_spread +
        0.1 * lex_order_weight
    )

    return final_priority
```

Figure 13: Priority function that constructs quaternary single-edit-correcting codes with minimum normalized gap to best-known sizes across all runs and search instances $n \in [6, 9]$.

```python
def priority(node, G, n, s):
    degree = G.degree[node]
    total_nodes = len(G.nodes())

    ham_distances = [
        sum(b1 != b2 for b1, b2 in zip(node, nb))
        for nb in G.neighbors(node)
    ]
    avg_ham = sum(ham_distances) / len(ham_distances) if ham_distances else 0
    var_ham = sum((hd - avg_ham)**2 for hd in ham_distances) / len(ham_distances) if
    ham_distances else 0

    bits = map(int, node)
    ones = sum(bits)
    probs = [ones/n, (n - ones)/n] if n > 0 else [0.5, 0.5]
    info_content = -sum(p * np.log2(p + 1e-6) for p in probs)

    def calc_seq_diversity():
        from itertools import combinations
        seen_seqs = set()
        depth = min(s, 3)

        for k in range(1, depth + 1):
            for indices in combinations(range(n), k):
                seq = ''.join(node[i] for i in range(n) if i not in indices)
                seen_seqs.add(seq)

        return len(seen_seqs) / (2**(depth + 1))

    seq_diversity = calc_seq_diversity()

    cover_degrees = [G.degree[nb] for nb in G.neighbors(node)] if G.neighbors(node) else []
    mean_cover_deg = sum(cover_degrees) / len(cover_degrees) if cover_degrees else 0
    influence_factor = 1/(mean_cover_deg + 1)

    final_score = (
        (-degree / total_nodes) *
        (seq_diversity ** 3.2) *
        (influence_factor ** 1.4) *
        (info_content + 0.6) *
        (var_ham + 1) ** 0.4 *
        (abs(avg_ham - n/2) ** 0.7)
    )

    return final_score
```

Figure 14: Priority function that constructs two-deletion-correcting codes with minimum normalized gap to best-known sizes across all runs and search instances $n \in [7, 12]$.

```python
def priority(node, G, n, s):
    degree = G.degree.get(node, 0)
    value = int(node, 2)
    hamming_weight = bin(value).count('1')
    density_ratio = hamming_weight / n
    connectivity_penalty = -math.log(max(degree, 1))
    if 0 < density_ratio < 1:
        entropy = -(density_ratio * math.log2(density_ratio) +
                    (1 - density_ratio) * math.log2(1 - density_ratio))
    else:
        entropy = 0
    target_lengths = (max(1, s - 5), min(n, s + 5))
    unique_samples = set()
    total_segments = 0
    for frag_len in range(*target_lengths):
        step_increment = 1 if frag_len <= 4 else 2
        for i in range(step_increment, n - frag_len + 1, step_increment):
            seg = node[i:i+frag_len]
            unique_samples.add(seg)
            total_segments += 1
    focal_feature_richness = (len(unique_samples) / total_segments if total_segments > 0
    else 0)
    balance_scores = []
    for span in [1, 2, 3]:
        chunks_available = n // span
        mirrored_sections = 0
        for chnk_num in range(chunks_available):
            front_segment = node[chnk_num*span:(chnk_num+1)*span]
            end_mirror = node[-(chnk_num+1)*span:-chnk_num*span]
            if front_segment == end_mirror:
                mirrored_sections += 1
        balance_scores.append(mirrored_sections / chunks_available)
    edge_matchings = (sum(1 for k in range(min(5, n // 2))
                         if node[k] == node[n-1-k]) /
                     min(5, n // 2))
    balance_scores.append(edge_matchings)
    architectural_stability = float(np.mean(balance_scores))
    disparities = []
    center_val = value
    for nbhr_node in G.adj[node]:
        diff_bits = bin(center_val ^ int(nbhr_node, 2)).count('1')
        disparities.append(diff_bits)
    if not disparities:
        heterogenetic_influence = 0
    else:
        std_deviations = float(np.std(disparities))
        norm_std = std_deviations / (math.sqrt(n) + 1e-6)
        heterogenetic_influence = 1 / (norm_std + 1e-6)
    unit_counts = {}
    positions = None
    for wdth in (2, 3):
        positions = list(range(n - wdth + 1))
        for pos in positions:
            substring = node[pos:pos+wdth]
            unit_counts[substring] = unit_counts.get(substring, 0) + 1
    if not unit_counts:
        recurrent_element_signature = 0
    else:
        most_popular = max(unit_counts.values())
        total_positions = len(positions) * 2
        recurrent_element_signature = most_popular / total_positions
    return (connectivity_penalty * 0.24 + entropy * 0.25 + focal_feature_richness * 0.32 +
    architectural_stability * 0.13 + heterogenetic_influence * 0.04 +
    recurrent_element_signature * 0.01)
```

Figure 15: Priority function that constructs three-deletion-correcting codes matching best-known sizes obtained with general-purpose graph solvers on search instances $n \in [8, 13]$.

```python
def priority(node, G, n, s):
    neighbors = list(G.neighbors(node))
    deg = G.degree[node]
    ndegs = [G.degree[nbr] for nbr in neighbors]
    avg_ndeg = np.mean(ndegs) if ndegs else 0.0

    tsig = -np.log1p(deg + avg_ndeg) * (1 + 0.1 * avg_ndeg / (deg + 1e-6))

    idx = np.arange(n)
    bits = np.array([int(c) for c in node], dtype=float)
    weights = (idx + 1) ** 0.7 * (n - idx) ** 0.7 * (1 - np.abs(idx - n // 2) / n) ** 0.25
    posi = float(bits @ weights) / max(1, len(G.nodes()) ** 0.3)

    overlap = sum(
        len({node[i:i + s + 1] for i in range(n - s) if node[i:i + s + 1] == nei[i:i + s +
1]})
        for nei in neighbors
    )
    olra = 1 - overlap / max(len(neighbors), 1)
    covf = len(neighbors) / max(len(G), 1)

    window_len = s + 1
    seen = set()
    for i in range(n - window_len + 1):
        pattern = node[i:i + window_len]
        for offset in (-1, 0, 1):
            seen.add((max(0, i + offset), pattern, offset))
    uc = len(seen) / max(n - window_len + 1, 1)

    return -(tsig * 0.25 + posi * 0.2 + covf * 0.15 + uc * 0.1 + olra * 0.05)
```

Figure 16: Priority function jointly optimized for single- and two-deletion-correcting codes and evaluated for single-, two-, and three-deletion correction.

Table 6: Statistics for the graphs used in our experiments. Each graph has $q^n$ nodes corresponding to all sequences of length $n$ over an alphabet of size $q$, with edges between pairs of sequences at Levenshtein distance $\leq 2s$ (deletion codes) or edit distance $\leq 2$ (insertion-deletion-substitution codes). Density is the ratio of edges to the maximum possible number of edges.

| | Binary single-deletion ($s = 1$, $q = 2$) | | | | | Quaternary edit ($s = 1$, $q = 4$) | | | |
|---|---|---|---|---|---|---|---|---|---|
| $n$ | Nodes | Edges | Avg. deg. | Den. (%) | $n$ | Nodes | Edges | Avg. deg. | Den. (%) |
| 6 | 64 | 543 | 17 | 26.9 | 6 | 4 096 | 392 358 | 192 | 4.7 |
| 7 | 128 | 1 471 | 23 | 18.1 | 7 | 16 384 | 2 206 374 | 269 | 1.6 |
| 8 | 256 | 3 839 | 30 | 11.8 | 8 | 65 536 | 11 815 590 | 361 | 0.55 |
| 9 | 512 | 9 727 | 38 | 7.4 | 9 | 262 144 | 60 992 166 | 465 | 0.18 |
| 10 | 1 024 | 24 063 | 47 | 4.6 | 10 | 1 048 576 | 305 965 734 | 584 | 0.056 |
| 12 | 4 096 | 139 263 | 68 | 1.7 | 11 | 4 194 304 | 1 500 162 726 | 715 | 0.017 |
| 14 | 16 384 | 761 855 | 93 | 0.57 | | | | | |
| 16 | 65 536 | 3 997 695 | 122 | 0.19 | | | | | |
| 18 | 262 144 | 20 316 159 | 155 | 0.059 | | | | | |
| 20 | 1 048 576 | 100 663 295 | 192 | 0.018 | | | | | |

| | Binary 2-deletion ($s = 2$, $q = 2$) | | | | | Binary 3-deletion ($s = 3$, $q = 2$) | | | |
|---|---|---|---|---|---|---|---|---|---|
| $n$ | Nodes | Edges | Avg. deg. | Den. (%) | $n$ | Nodes | Edges | Avg. deg. | Den. (%) |
| 6 | 64 | 1 494 | 47 | 74.1 | 6 | 64 | 1 906 | 60 | 94.5 |
| 7 | 128 | 5 173 | 81 | 63.6 | 7 | 128 | 7 361 | 115 | 90.6 |
| 8 | 256 | 17 183 | 134 | 52.6 | 8 | 256 | 27 868 | 218 | 85.4 |
| 9 | 512 | 54 895 | 214 | 42.0 | 9 | 512 | 103 272 | 403 | 78.9 |
| 10 | 1 024 | 169 162 | 330 | 32.3 | 10 | 1 024 | 373 784 | 730 | 71.4 |
| 12 | 4 096 | 1 460 525 | 713 | 17.4 | 12 | 4 096 | 4 531 232 | 2 213 | 54.0 |
| 14 | 16 384 | 11 342 596 | 1 385 | 8.5 | 14 | 16 384 | 49 323 673 | 6 021 | 36.8 |
| 16 | 65 536 | 80 964 349 | 2 471 | 3.8 | 16 | 65 536 | 484 191 170 | 14 776 | 22.5 |
| 18 | 262 144 | 540 270 870 | 4 122 | 1.6 | 18 | 262 144 | 4 331 830 631 | 33 049 | 12.6 |

# G   Graph statistics

Table 6 summarizes graph statistics for all problem instances we consider. For one and two deletions, the graphs become sparse at longer code lengths and kernelization-based MIS solvers from KaMIS find competitive solutions. For three deletions, the graphs are still dense even at longer code lengths (e.g., 12.6% at code length 18 with over 4.3 billion edges) and KaMIS times out without finding a solution.

