# OpenReview forum: "LLM-Guided Search for Deletion-Correcting Codes"
_TMLR — Accepted by TMLR_

### Review · Reviewer_mFoM · 2026-04-04

**Summary Of Contributions:**

This paper shows how to use LLM-guided evolutionary search to automatically discover better deletion-correcting codes. The paper explains the problem and how to sovle it. It uses python functions to illustrate the process. The paper includes empirical results.

**Audience:**

No

**Audience Explanation:**

Although the paper includes LLM and deep learning teminologies, the paper is better submitted to the information theory area journals. The paper has no comparisons with other coding methods, and the detail application descriptions are not found in the paper.

**Broader Impact Concerns:**

No.

**Claims And Evidence:**

No

**Claims Explanation:**

No new theoretial insight. Results are purely empirical. No closed-form construction is provided. The experimental settings are small and have high variance. The code length is short in the paper.

**Requested Changes:**

Includes theoretical analysis. Compare wtih other coding methods. Submit to information theory related journals.

---

> ### Author Response · Authors · 2026-05-03
>
> We address each of the concerns below.
>
> **Regarding theoretical analysis:** In Appendix D, we analyze the priority function in Figure 3 theoretically and prove that combined with greedy construction and lexicographic tie-breaking, it constructs the largest $\mathrm{VT}_0(n)$ code for any code length $n$. The proof shows that for any sequence outside $\mathrm{VT}_0(n)$, there exists a sequence inside $\mathrm{VT}_0(n)$ that shares a length-$(n-1)$ subsequence with it and has either strictly higher priority, or the same priority and smaller lexicographic order, so the greedy algorithm always selects the $\mathrm{VT}_0(n)$ sequence first. More generally, each priority function, combined with the greedy construction, gives an explicit and deterministic way to construct a code at any length and deletion count.
>
> The theoretical analysis in Appendix D also shows an advantage of LLM-guided search over searching in the sequence space or general-purpose graph solvers as the discovered functions can be analyzed, and in some cases their resulting code sizes can be characterized without explicit enumeration at each length.
>
> **Regarding comparisons with other coding methods:** Tables 1-3 compare against what we believe are all relevant baselines for the finite-length regime we consider: explicit constructions from coding theory [1, 2], search-based methods over sequence space [3], the recent DoDo-Code neural method [4], general-purpose graph solvers (OnlineMIS and ReduMIS from KaMIS [5, 6], and Gurobi as an ILP solver), random greedy construction, and LP/ILP upper bounds. If the reviewer has a specific method in mind that we missed, we would be happy to include it.
>
> We do not compare against asymptotically optimal constructions for multiple deletions, e.g., [7-12], as comparing them in the finite-length regime would be misleading. Even ignoring the lower-order terms, which are non-negligible at moderate code lengths, the leading-term redundancy of these constructions is already larger or close to the codeword length itself in our regime. For example, the best known asymptotic construction for two deletions [12] achieves redundancy $4 \log_2 n + O(\log \log n)$, which gives $4 \log_2(18) \approx 16.7$ bits redundancy at $n=18$ (the largest length we consider) and a code of size $2^{1.3} \approx 2.5$, compared to a code size of 539 achieved by one of our priority functions.
>
> **Regarding short code lengths:** We indeed consider short code lengths. However, short codes can also be practically relevant, since asymptotic constructions for multiple deletions [7-12] partition codewords into short blocks and use short deletion-correcting codes to protect the block contents. To construct these short inner codes, they use a brute-force greedy coloring of the confusability graph, which produces code sizes equivalent to iteratively adding sequences in the order of a random permutation, i.e., our random-greedy baseline. Since our discovered priority functions construct larger codebooks than this baseline (Tables 1-3), they could in principle be substituted for the brute-force inner code in these asymptotic constructions, reducing redundancy at moderate lengths (e.g., 100-200, where constants still matter). We have added this discussion to the revised Section 2.2.
>
> **Regarding experimental variance:** While we only report three seeds per configuration due to computational cost, we believe our findings are robust for three reasons. First, for the single-deletion ablations in Section 5.1, the min-max bands for the top-50 mean gap (Figure 4, left) are small for all configurations, suggesting low variance across seeds. Second, for the two-deletion ablations in Section 5.2.1, the same trend, that co-evolving an algorithmic summary with code hurts performance, holds across all three Qwen3 model sizes (8B, 14B, 32B). To further strengthen this finding, in the revision we additionally evaluate the Gemma 4 family (E4B-it, 26B-A4B-it, 31B-it) and find the same trend (Figure 8, Appendix E). Third, for the reasoning-mode ablation, Figure 6 reports results averaged over the 4 prompting strategies, giving 12 runs per mode and model, and shows that for our problem reasoning does not improve performance under a fixed inference budget across all three Qwen3 model sizes. We confirm this trend on the Gemma 4 family as well in the revision (Figure 9, Appendix E).

---

> > ### Author Response · Authors · 2026-05-03
> >
> > [1] Varshamov, R. R., & Tenengolts, G. M. (1965). Codes which correct single asymmetric errors (in Russian). Automatika i Telemkhanika, 161(3), 288-292.
> >
> > [2] Helberg, A. S., & Ferreira, H. C. (2002). On multiple insertion/deletion correcting codes. IEEE Transactions on Information Theory, 48(1), 305-308.
> >
> > [3] Landjev, I., & Haralambiev, K. (2007). On multiple deletion codes. Serdica Journal of Computing, 1(1), 13-26.
> >
> > [4] Guo, A. J., Sun, S., Wei, X., Wei, M., & Chen, X. (2023). DoDo-Code: an Efficient Levenshtein Distance Embedding-based Code for 4-ary IDS Channel. arXiv preprint arXiv:2312.12717.
> >
> > [5] Dahlum, J., Lamm, S., Sanders, P., Schulz, C., Strash, D., & Werneck, R. F. (2016, June). Accelerating local search for the maximum independent set problem. In International symposium on experimental algorithms (pp. 118-133). Cham: Springer International Publishing.
> >
> > [6] Lamm, S., Sanders, P., Schulz, C., Strash, D., & Werneck, R. F. (2016). Finding near-optimal independent sets at scale. In 2016 Proceedings of the eighteenth workshop on algorithm engineering and experiments (ALENEX) (pp. 138-150). Society for Industrial and Applied Mathematics.
> >
> > [7] Brakensiek, J., Guruswami, V., & Zbarsky, S. (2017). Efficient low-redundancy codes for correcting multiple deletions. IEEE Transactions on Information Theory, 64(5), 3403-3410.
> >
> > [8] Gabrys, R., & Sala, F. (2018). Codes correcting two deletions. IEEE Transactions on Information Theory, 65(2), 965-974.
> >
> > [9] Sima, J., Gabrys, R., & Bruck, J. (2020, June). Optimal systematic t-deletion correcting codes. In 2020 IEEE international symposium on information theory (ISIT) (pp. 769-774). IEEE.
> >
> > [10] Sima, J., Raviv, N., & Bruck, J. (2019). Two deletion correcting codes from indicator vectors. IEEE transactions on information theory, 66(4), 2375-2391.
> >
> > [11] Sima, J., & Bruck, J. (2020). On optimal k-deletion correcting codes. IEEE Transactions on Information Theory, 67(6), 3360-3375.
> >
> > [12] Guruswami, V., & Håstad, J. (2021). Explicit two-deletion codes with redundancy matching the existential bound. IEEE Transactions on Information Theory, 67(10), 6384-6394.

---

### Review · Reviewer_BLNE · 2026-04-16

**Summary Of Contributions:**

This paper adapts FunSearch, an LLM guided evolutionary search framework, to deletion-correcting code construction. The authors reduce code design to a maximum independent set problem on a sequence graph, and use an LLM to evolve Python priority functions that drive a greedy code construction process. For the single-deletion setting, the method rediscovers a construction provably equivalent to the optimal VT code. The paper also provides empirical evidence that, for this type of algorithmic discovery, allocating compute to broader sampling is more effective than spending more budget on deeper reasoning per sample.

Strengths:
1. The discovery of a priority function provably equivalent to the VT code gives the paper an unusual level of rigor for LLM-guided search work.
2. Output-level logical deduplication is simple but effective, and appears crucial for maintaining diversity during search.

Weaknesses:
1. The evaluator scales exponentially with code length, making the current approach impractical beyond short or moderate regimes.
2. The poor performance of larger models suggests that the framework is still brittle with respect to model choice and generation behavior.

**Audience:**

Yes

**Audience Explanation:**

The paper addresses a classical and difficult problem in coding theory and combines it with a currently important methodological question, whether LLM-guided search can meaningfully contribute to scientific or algorithmic discovery. These results are relevant not only to researchers interested in LLM-based discovery systems, but also to readers working on combinatorial optimization, coding theory, and automated heuristic design.

**Broader Impact Concerns:**

I did not identify any severe ethical risks that would by themselves require an extensive broader impact discussion. The work is primarily methodological and focuses on code construction in information theory, with no obvious direct misuse pathway or harmful deployment scenario discussed in the paper.

**Claims And Evidence:**

Yes

**Claims Explanation:**

The paper provides strong support for some of its claims, especially in the single-deletion setting, where one discovered function is proven equivalent to the VT construction, and in the ablations showing the importance of deduplication. However, some broader conclusions are overstated. In particular, the claim that larger models are less effective is heavily confounded by the catastrophic execution failure rate of the specific 32B model used, and the empirical study remains confined to short finite-length regimes due to the exponential evaluator bottleneck. Thus, while the paper contains interesting and often convincing evidence within its tested setting, it does not yet fully justify all of its broader claims.

**Requested Changes:**

Critical:
1. The paper does not currently justify the broad conclusion that “larger models are less effective.” In practice, this claim is heavily confounded by the catastrophic execution failure rate of Qwen3-32B (>95%), suggesting that the observed weakness may reflect a model–prompt–framework mismatch rather than an intrinsic disadvantage of larger models. The authors should either investigate and mitigate these formatting/execution failures, or substantially narrow the claim to the specific models and setup studied here.
2. Because the evaluator relies on explicit sequence enumeration and graph construction, the method scales exponentially and the empirical results are confined to very short code lengths. This is a major limitation, not a minor caveat. The Abstract and Introduction should explicitly state that the current approach is only demonstrated in short-length regimes and should not be interpreted as immediately applicable to larger or practically relevant code constructions.

Strengthening:
1. To strengthen the finding that "sampling outperforms reasoning," the authors should test at least one alternative reasoning model family (e.g., DeepSeek) to prove this conclusion is robust to the problem structure and not merely a limitation of the Qwen3 series.

---

> ### Author Response · Authors · 2026-05-03
>
> We thank the reviewer for their feedback and for the positive comments on our theoretical result and on the role of deduplication for search diversity. We address each concern below.
>
> **Regarding the claim that larger models are less effective:** We agree the original claim was too broad. We have revisited the choice of hyperparameters for Qwen3-32B to address the high execution failure rate. In our original setup, we used the same hyperparameters (i.e., temperature = 0.94, top_p = 0.78, repetition_penalty = 1.2 and no top-k filtering) for StarCoder2-15B and across all Qwen3 sizes. The outputs of Qwen3-32B looked sensible, and we initially thought the high failure rate came from the larger model attempting more complex code with more room for syntax errors and timeouts. We reran the experiments using the recommended hyperparameters from the Qwen3 model card for both non-thinking and thinking modes, which reduced the failure rate from above 95% to about 20%, and all Qwen3 sizes now perform similarly. We also ran small-scale experiments for Qwen3-8B and Qwen3-14B ($20 per prompting strategy, with and without reasoning) and found the recommended hyperparameters performed similarly to or slightly worse than our original choice, so we kept the original setting for these two models. For Qwen3-8B, we additionally noticed that our original run had used the sampling period (P=30K) and reset threshold (R=1.2K) from our StarCoder2-15B setup, rather than the updated values (P=180K, R=6K) we use for the Qwen3 and Gemma-4 families to account for their lower execution failure rates. The updated curves (now Figure 5, previously Figure 7 in Appendix F) use the corrected sampling period and reset threshold, and our conclusions are unchanged.
> We additionally ran experiments on the Gemma 4 family (E4B, 26B-A4B, 31B) in Appendix E, and find the same pattern, with all model sizes performing comparably. Based on these results, we removed the claim that larger models are less effective and the corresponding figure. However, we think it is still notable that larger models are not more effective for our task than smaller ones.
>
> **Regarding scalability and short code lengths:** We agree that the exponential scaling of the evaluator is an important limitation. We have updated the Abstract and Introduction to explicitly state that we focus on the short length regime because code construction scales exponentially with code length. We have also added a section on scalability limits (Section 6 in the revised paper), which was previously only discussed together with the conclusion.
> However, our functions can in principle be used at longer code lengths as inner codes in asymptotic constructions [1-4] that split codewords into short blocks and use short deletion-correcting codes to protect each block. The short inner codes are typically built using a brute-force greedy coloring of the confusability graph, which gives the same code sizes as adding sequences in the order of a random permutation (our random-greedy baseline in Tables 1-3). Since our discovered priority functions construct larger codebooks than random greedy at the lengths we consider, they could replace the brute-force inner code (see also revised Section 2.2).
>
> **Strengthening the finding that sampling outperforms reasoning:** To strengthen our finding that sampling outperforms reasoning for our problem, we additionally ran experiments with the Gemma 4 model family, which similarly supports toggling reasoning on and off within the same model. We chose Gemma 4 over DeepSeek because DeepSeek does not support toggling reasoning within a single model. We find that also for Gemma 4 sampling more functions is more effective than extended reasoning per function under a fixed inference budget for our problem. The table below shows the top-50 mean gap, averaged across the code, reflection, and co-evolving prompting strategies, with 3 seeds per strategy. The full plot is included in Appendix E of the revised paper.
>
> | | gemma-4-E4B-it |gemma-4-26B-A4B-it | gemma-4-31B-it |
> |---|---|---|---|
> | No reasoning | 0.0759 ± 0.0054 | 0.0711 ± 0.0074 | 0.0688 ± 0.0037 |
> | Reasoning | 0.0847 ± 0.0039 | 0.0966 ± 0.0038 | 0.0907 ± 0.0119 |
>
> We hope this addresses the reviewer's concerns and would be happy to clarify any remaining points.

---

> > ### Author Response · Authors · 2026-05-03
> >
> > [1] Gabrys, R., & Sala, F. (2018). Codes correcting two deletions. IEEE Transactions on Information Theory, 65(2), 965-974.
> >
> > [2] Sima, J., Gabrys, R., & Bruck, J. (2020, June). Optimal systematic t-deletion correcting codes. In 2020 IEEE international symposium on information theory (ISIT) (pp. 769-774). IEEE.
> >
> > [3] Sima, J., Raviv, N., & Bruck, J. (2019). Two deletion correcting codes from indicator vectors. IEEE transactions on information theory, 66(4), 2375-2391.
> >
> > [4] Guruswami, V., & Håstad, J. (2021). Explicit two-deletion codes with redundancy matching the existential bound. IEEE Transactions on Information Theory, 67(10), 6384-6394.

---

### Review · Reviewer_hDb8 · 2026-04-20

**Summary Of Contributions:**

This paper adapts FunSearch-style LLM-guided evolutionary search to construct deletion-correcting codes by searching over priority functions for greedy independent set construction. For single-deletion binary codes, the method rediscovers functions equivalent to VT codes, including one that is proven equivalent to VT_{0}(n) for all n. For multiple deletions and quaternary edit codes, the discovered functions outperform several explicit or heuristic baselines, though not the best general-purpose graph solvers. The paper also contributes an empirical study of search design choices, finding that output-level deduplication is critical, co-evolving descriptions is unhelpful, and under a fixed budget it is better to sample more functions than spend more compute per function.

Strengths: strong empirical study, clear problem formulation, useful ablations, and an interesting negative result about reasoning-vs-sampling. Weaknesses: limited scalability, modest gains beyond rediscovering known constructions in the strongest setting, and somewhat broad claims about “scientific discovery” relative to the actual coding-theoretic progress.

**Audience:**

Yes

**Audience Explanation:**

I think this will interest readers working on LLMs for scientific discovery, automated heuristic design, and combinatorial search. The most interesting takeaway is not just the application to coding theory, but the careful evidence that, for this domain, breadth of search matters more than deeper reasoning per sample. That is a useful result for the broader LLM-guided search literature. The paper is also a good cross-disciplinary example connecting ML search methods with information/coding theory.

**Broader Impact Concerns:**

I do not see major ethical concerns.

**Claims And Evidence:**

Yes

**Claims Explanation:**

Mostly yes. The empirical evidence is solid for the paper’s main methodological claims. The authors compare against explicit constructions, search heuristics, random greedy baselines, and MIS/ILP solvers, and they provide useful ablations on deduplication, graph input, scoring, prompting, model choice, and reasoning budget. Tables 1–3 and the ablation figures support the core conclusions well.

**Requested Changes:**

- The paper should distinguish more clearly between rediscovering known optimal structure and finding genuinely new coding-theoretic insights.
- Expand the discussion of scalability limits. Since the evaluator is the main bottleneck, this deserves a more prominent discussion in the main text.

---

> ### Author Response · Authors · 2026-05-03
>
> We thank the reviewer for their feedback and for the positive comments on our ablations and the reasoning vs. sampling finding. We address each of the requested changes below.
>
> **Regarding distinguishing rediscovery from new coding-theoretic insights:** We agree with the reviewer and have revised the paper to make this distinction clearer. Specifically:
>
> - In the Abstract, we changed "our search discovers a construction provably equivalent to the conjectured-optimal VT code" to "our search finds a function that we prove constructs the conjectured-optimal VT code," separating the search output (a function) from the construction (a known result). In the first contributions bullet (Introduction), we added "rediscovering a known optimal construction" after the single-deletion claim and in Section 5.1, we revised the memorization paragraph from "we argue this is a genuine discovery rather than memorization, as the discovered functions use different construction logic than VT codes typically use" to "we argue this is a genuine rediscovery rather than memorization, as the search did not generate their standard modular-sum form." We hope these edits better distinguish that the function in Figure 3 rediscovered a known construction.
>
> - In Section 5.1, we replaced "novel constructions" with "alternative constructions" when describing the codes in Figure 11, which match optimal sizes at $n \in \lbrace 6, 7, 9, 10, 11 \rbrace$ but are distinct from all $\mathrm{VT}_a(n)$ codes. This avoids implying a new structural finding, since the non-uniqueness of maximum-size single-deletion codes at small lengths is itself known.
>
> - For multiple binary deletions and quaternary edit codes, the discovered functions combine heuristics (low degree, small deletion ball, low binary entropy, Hamming distance statistics) with learned weightings and we could not yet find new coding-theoretic insights (we analyze a representative function in Appendix F). To reflect this, we added a closing sentence to the first contributions bullet, "[...] analyzing them to derive new coding-theoretic insights or efficient encoders remains an open problem", and in the conclusion changed "finding new codes for multiple binary deletions and quaternary edit correction'" to "constructing larger codes than prior search-based and explicit constructions for multiple binary deletions and quaternary edit correction'". Together, we hope these edits make clear that the search produces larger codes but not yet new structural insights for multiple deletions.
>
> - In the caption of Figure 10, we added the simplified form of the priority, $\sum_{1 \leq i \leq j \leq n} \bigl(\sum_{k=i}^{j} 1[x_k = 0]\bigr)^2$, to make explicit that the function uses a quadratic sum on zero positions rather than the linear sum that VT codes use, while still constructing the $\mathrm{VT}_0(n)$ codebook at even lengths and its complement at odd lengths (verified for $n \in [6, 25]$). Since VT codes are often used as a building block in multiple-deletion-correcting codes [1], we tested whether our priority function offers any advantage over VT codes for two deletions. We applied our function with the greedy algorithm at lengths $n \in [7, 12]$ and compared against the largest two-deletion-correcting subset of $\mathrm{VT}_0(n)$, computed by exact MIS on the $\mathrm{VT}_0$ subgraph of the two-deletion confusability graph. Our priority function constructs larger codes at all tested lengths except $n=7$:
>
>   |                                 | n=7 | n=8 | n=9 | n=10 | n=11 | n=12 |
>   |---------------------------------|----:|----:|----:|-----:|-----:|-----:|
>   | Our function                    |   4 |   7 |   9 |   13 |   20 |   33 |
>   | Exact MIS on $\mathrm{VT}_0(n)$ |   4 |   5 |   7 |   12 |   17 |   26 |
>
>   We have not yet tested this at larger lengths or studied how our priority function interacts with other sketch functions used in multi-deletion code constructions, but view it as an interesting new insight and starting point.
>
> [1] Guruswami, V., & Håstad, J. (2021). Explicit two-deletion codes with redundancy matching the existential bound. IEEE Transactions on Information Theory, 67(10), 6384-6394.

---

> > ### Author Response · Authors · 2026-05-03
> >
> > **Regarding the discussion of scalability limits.** We have added a section in the main text (Section 6) discussing scalability in more detail, which we previously only discussed together with the conclusion. Specifically, we have added the following paragraphs:
> >
> >  "A key limitation of our approach is the scalability of the evaluator, which makes using our priority functions at larger code lengths infeasible. Our greedy construction algorithm requires computing priorities for all sequences and therefore scales exponentially in code length.
> >
> >   To use our priority functions at larger code lengths, we would need an efficient encoder that maps a message to a codeword without enumerating all sequences. Constructing such an encoder requires (1) characterizing which priority scores belong to the codebook for any length $n$, and (2) determining how to insert redundant bits to achieve a target priority score. We demonstrate (1) for the function in Figure 3 (Appendix D) where (2) follows from the known $\mathrm{VT}_0(n)$ encoder. Deriving similar characterizations or encoders for our multi-deletion functions remains an open problem.
> >
> >   Nevertheless, compared to previous search-based methods, the discovered priority functions can be analyzed mathematically, which is harder for codes found by direct search. An advantage over general-purpose graph solvers is that evaluating our functions does not require the full confusability graph, which quickly becomes memory-infeasible. As long as the priority function does not depend on global graph statistics (which is the case for our best discovered functions), priority computation per sequence is polynomial in $n$."
> >
> > We hope this addresses the reviewer’s concern and are happy to clarify any remaining points further.

---

### Decision · Action_Editor_GynE · 2026-05-24

**Recommendation:** Accept with minor revision

**Additional Comments:**

The authors have addressed the major reviewer concerns in the rebuttal and revision. In the final version, they should ensure that the clarified positioning is consistently reflected throughout the paper, particularly regarding the scalability limitations, the scope of conclusions about model size and reasoning, and the distinction between rediscovering known constructions and obtaining new coding-theoretic insights. The paper should also clearly emphasize that the multi-deletion results remain primarily empirical and heuristic, with limited theoretical understanding beyond the demonstrated finite-length improvements.

**Audience:**

Yes

**Audience Explanation:**

The paper’s combination of LLM-guided discovery, coding theory, and empirical insights on search strategies should interest readers working on scientific discovery and automated algorithm design.

**Claims And Evidence:**

Yes

**Claims Explanation:**

The paper provides convincing empirical evidence for its core claims through strong baselines, ablations, and the rediscovery of a construction provably equivalent to the VT code. The rebuttal and revision also substantially clarified previously overstated claims regarding scalability and larger-model effectiveness. Overall, the claims are well supported within the demonstrated scope.